# Scattering Vision Transformer: Spectral Mixing Matters

**Badri Narayana Patro**
Microsoft
badripatro@microsoft.com

Vijay Srinivas Agneeswaran
Microsoft
vagneeswaran@microsoft.com

## Abstract

Vision transformers have gained significant attention and achieved state-of-the-art performance in various computer vision tasks, including image classification, instance segmentation, and object detection. However, challenges remain in addressing attention complexity and effectively capturing fine-grained information within images. Existing solutions often resort to down-sampling operations, such as pooling, to reduce computational cost. Unfortunately, such operations are non-invertible and can result in information loss. In this paper, we present a novel approach called Scattering Vision Transformer (SVT) to tackle these challenges. SVT incorporates a spectrally scattering network that enables the capture of intricate image details. SVT overcomes the invertibility issue associated with down-sampling operations by separating low-frequency and high-frequency components. Furthermore, SVT introduces a unique spectral gating network utilizing Einstein multiplication for token and channel mixing, effectively reducing complexity. We show that SVT achieves state-of-the-art performance on the ImageNet dataset with a significant reduction in a number of parameters and FLOPS. SVT shows 2% improvement over LiTv2 and iFormer. SVT-H-S reaches 84.2% top-1 accuracy, while SVT-H-B reaches 85.2% (state-of-art for base versions) and SVT-H-L reaches 85.7% (again state-of-art for large versions). SVT also shows comparable results in other vision tasks such as instance segmentation. SVT also outperforms other transformers in transfer learning on standard datasets such as CIFAR10, CIFAR100, Oxford Flower, and Stanford Car datasets. The project page is available on this webpage (https://badripatro.github.io/svt/).

## 1 Introduction

In recent years, there has been a remarkable surge in the interest and adoption of Large Language Models (LLMs), driven by the release and success of prominent models such as GPT-3, ChatGPT [1], and Palm [8]. These LLMs have achieved significant breakthroughs in the field of Natural Language Processing (NLP). Building upon their successes, subsequent research endeavors have extended the language transformer paradigm to diverse domains including computer vision, speech recognition, video processing, and even climate and weather prediction. In this paper, we specifically focus on exploring the potential of LLMs for vision-related tasks. By leveraging the power of these language models, we aim to push the boundaries of vision applications and investigate their capabilities in addressing complex vision challenges.

Several adaptations of transformers have been introduced in the field of computer vision for various tasks. For image classification, notable vision transformers include ViT [13], DeIT [56], PVT [61], Swin [39], Twin [9], and CSWin transformers [12]. The different vision transformers improved the performance of image classification tasks significantly compared to Convolutional Neural Networks (CNNs) such as ResNets and RegNets, as discussed in efficient vision transformer research work [43]. This breakthrough in computer vision has led to state-of-the-art results in various vision tasks, including image segmentation such as SegFormer [66], TopFormer [77] and SegViT [74] and object

37th Conference on Neural Information Processing Systems (NeurIPS 2023).

detection, models like DETR [4] and Yolo[15]. However, one challenge faced by vision transformers is the increasing computational complexity of the self-attention module as the sequence length or image resolution grows. Additionally, model size and the number of floating-point operations per second (FLOPS) also increase with image resolution or sequence length. These factors need to be carefully considered and addressed to ensure efficient and scalable deployment of vision transformers in real-world applications.

One way to address the computational complexity of attention-based Transformers is to replace the attention mechanism with a Multi-Layer Perceptron (MLP) based mixer layer [54, 55, 18]. However, it is difficult to capture spatial information in the MLP mixers. This was addressed by the paper MetaFormer [71]. MetaFormer uses a pooling operation to replace the attention layer. This however has the disadvantage that the pooling operation is not invertible and could possibly lose information. Fourier based Transformers such as FourierFormer[53], FNet[31], GFNet [47] and AFNO [16] minimizes the loss of information by using Fourier Transform. But it has an inherent problem of separating the low and high-frequency components. The ability to separate low-frequency and high-frequency components of an image is important. Recently, transformers such as LiTv2 [42] and iFormer [51] have been proposed to address this problem. However, both LiTv2 and iFormer have the same $O(n^2)$ complexity as they use full-fledged self-attention networks, similar to ViT [13] and DeIT [56], which are weak in capturing fine-grained information of images. Vit, DeIT, LiTv2 and iFormer also have a limitation with respect to network size or number of parameters. We propose a Scattering Vision Transformer (SVT) which uses a spectral scattering network to address the attention complexity and Dual-Tree Complex Wavelet Transform (DTCWT) to capture the fine-grained information using spectral decomposition into low-frequency and high-frequency components of an image.

SVT uses the scattering network as the initial layer of the transformer, which captures fine-grained information (lines and edges, for instance) along with the energy component. SVT also uses attention nets in the deeper layers to capture long-range dependencies. The fine-grained information is captured by the high-frequency component of the scattering network by using the DTCWT transform while the low-frequency component is the energy component. SVT uses a Spectral Gating Network (SGN) to capture the effective features in both frequency components. Generally, the high-frequency component has extra directional information which makes it computationally complex while performing the gating operation. SVT addresses this complexity with a novel token and channel mixing technique using the Einstein Blending Method (EBM) in high-frequency component. SVT also uses a Tensor Blending Method (TBM) in the low-frequency component. We also observe that the DTCWT is more invertible compared to other spectral transformations in the literature which are based on Fourier transforms and discrete wavelet transforms [2, 28]. We quantify the invertibility in terms of reconstruction loss in the performance study section of this paper. The use of TBM for low-frequency components and EBM for high-frequency components is our contribution. It must be noted that low-frequency components contain the energy component of the signal which requires all the frequency components to provide energy compaction, while high-frequency components can be represented by only a few components, which can be achieved using Einstein multiplication. SVT is a generic recipe for componentizing the transformer architecture and efficiently implementing transformers with lesser parameters and computational complexity with the help of Einstein multiplication. So, this can be viewed as a simple and efficient learning transformer architecture with minimal inductive bias.

Our contributions are as follows:

- We introduce a novel invertible scattering network based on DTCWT transformation into vision transformers to decompose image features into low-frequency and high-frequency features.

- We proposed a novel SGN, which uses TBM to mix low-frequency representations and EBM to mix high-frequency representations. We use an efficient way of mixing high-frequency components using channel and token mixing with the help of Einstein multiplication.

- Detailed performance analysis shows that SVT outperforms all transformers including LiT v2 and iFormer on ImageNet data, with a significantly lesser number of parameters. In addition, SVT also has comparable performance on other transfer learning datasets.

- We show that SVT is efficient not only performance-wise but also in terms of a number of parameters (memory size) as well as in terms of computational complexity (measured in Gigaflops). We also show that SVT is efficient for inferencing, by measuring its latency and comparing it with other state-of-art transformers.

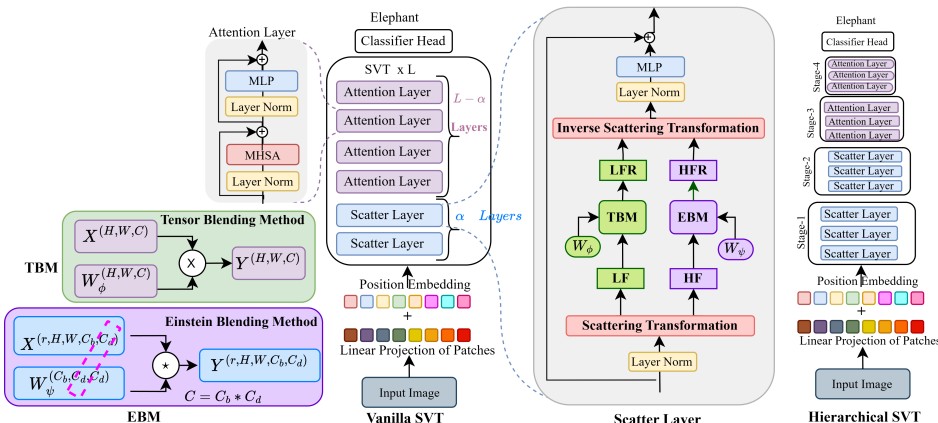

Figure 1: This figure illustrates the architectural details of the SVT model with a Scatter and Attention Layer structure. The Scatter Layer comprises a Scattering Transformation that processes Low-Frequency (LF) and High-Frequency (HF) components. Subsequently, we apply the Tensor and Einstein Blending Method to obtain Low-Frequency Representation (LFR) and High-Frequency Representation (HFR), as depicted in the figure. Finally, we apply the Inverse Scattering transformation using LFR and HFR.

## 2 Method

### 2.1 Background: Overview of DTCWT and Decoupling of Low & High Frequencies

Discrete Wavelet Transform (DWT) replaces the infinite oscillating sinusoidal functions with a set of locally oscillating basis functions, which are known as wavelets [50, 27]. Wavelet is a combination of low-pass scaling function $\phi(t)$ and a shifted version of a band-pass wavelet function known as $\psi(t)$. It can be represented mathematically as given below:

$$x(t) = \sum_{n=-\infty}^{\infty} c(n)\phi(t-n) + \sum_{j=0}^{\infty} \sum_{n=-\infty}^{\infty} d(j,n)2^{j/2}\psi(2^j t - n). \tag{1}$$

where $c(n)$ is the scaling coefficients and $d(j,n)$ is the wavelet coefficients. These coefficients are computed by the inner product of the scaling function $\phi(t)$ and wavelet function $\psi(t)$ with input $x(t)$.

$$c(n) = \int_{-\infty}^{\infty} x(t)\phi(t-n)dt, \quad d(j,n) = 2^{j/2} \int_{-\infty}^{\infty} x(t)\psi(2^j t - n)dt. \tag{2}$$

DWT suffers from the following issues oscillations, shift variance, aliasing, and lack of directionality. One of the solutions to solve the above problems is the Complex Wavelet Transform (CWT) with complex-valued scaling and wavelet function. The Dual-Tree Complex Wavelet Transform (DT-CWT) addresses the issues of the CWT. The DT-CWT [28, 26, 27] comes very close to mirroring the attractive properties of the Fourier Transform, including a smooth, nonoscillating magnitude, a nearly shift-invariant magnitude with a simple near-linear phase encoding of signal shifts, substantially reduced aliasing; and better directional selectivity wavelets in higher dimensions. This makes it easier to detect edges and orientational features of images. The six orientations of the wavelet transformation are given by 15°, 45°, 75°, 105°, 135°, and 165°. The dual-tree CWT employs two real DWTs, the first DWT gives the real part of the transform while the second DWT gives the imaginary part. The two real DWTs use two different sets of filters, which are jointly designed to give an approximation of the overall complex wavelet transform and satisfy the perfect reconstruction (PR) conditions.

Let $h_0(n), h_1(n)$ denote the low-pass and high-pass filter pair in the upper band, while $g_0(n), g_1(n)$ denote the same for the lower band. Two real wavelets are associated with each of the two real wavelet transforms as $\psi_h(t)$, and $\psi_g(t)$. The complex wavelet $\psi_h(t) := \psi_h(t) + \psi_g(t)$ can be approximated using Half-Sample Delay[49] condition,i.e. $\psi_h(t)$ is approximately the Hilbert transform of $\psi_g(t)$ like

$$g_0(n) \approx h_0(n-0.5) \Rightarrow \psi_g(t) \approx \mathcal{H}\{\psi_h(t)\}\psi_h(t) = \sqrt{2}\sum_n h_1(n)\phi_h(t), \phi_h(t) = \sqrt{2}\sum_n h_0(n)\phi_h(t)$$

Similarly, we can define $\psi_g(t), \phi_g(t),$ and $g_1(n)$. Since the filters are real, no complex arithmetic is required to implement DTCWT. It is just two times more expansive in 1D because the total output data rate is exactly twice the input data rate. It is also easy to invert, as the two separate DWTs can be inverted. Compare DTCWT with the Fourier Transform, which is difficult to obtain low pass and high pass components of an image and it is less invertible (Loss is high when we do Fourier and inverse Fourier transform) compared to DTCWT. Also, It cannot speak about time and frequency simultaneously.

## 2.2 Scattering Visual Transformer (SVT) Method

Given input image $\mathbf{I} \in \mathbb{R}^{3 \times 224 \times 224}$, we split the image into the patch of size $\mathbb{R}^{16 \times 16}$ and obtain embedding of each patch token using position encoder and token embedding network. $\mathbf{X} = \mathcal{F}_T(\mathbf{I}) + \mathcal{F}_P(\mathbf{I})$, where $\mathcal{F}_T, \mathcal{F}_P$ refer to token and position encoding network. The detailed distinct components of the SVT architecture are illustrated in Figure 1. Scattering Visual Transformer consists of three components such as a) Scattering Transformation, b) Spectral Gating Network, c) Spectral Channel and Token Mixing.

**A. Scattering Transformation:**

The input image $\mathbf{I}$ is firstly patchified into a feature tensor $\mathbf{X} \in \mathbb{R}^{C \times H \times W}$ whose spatial resolution is $H \times W$ and the number of channels is $C$. To extract the features of an image, we feed $\mathbf{X}$ into a series of transformer layers. We use a novel spectral transform based on an invertibility scattering network instead of the standard self-attention network. This allows us to capture both the fine-grain and the global information in the image. The fine-grain information consists of texture, patterns, and small features that are encoded by the high-frequency components of the spectral transform. The global information consists of the overall brightness, contrast, edges, and contours that are encoded by the low-frequency components of the spectral transform. Given feature $\mathbf{X} \in \mathbb{R}^{C \times H \times W}$, we use scattering transform using DTCWT [50] as discussed in section-2.1 to obtain the corresponding frequency representations $\mathbf{X}_F$ by $\mathbf{X}_F = \mathcal{F}_{\text{scatter}}(\mathbf{X})$. The transformation in frequency domain $\mathbf{X}_F$ provides two components, one low-frequency component i.e. scaling component $\mathbf{X}_\phi$, and one high-frequency component i.e. wavelet component $\mathbf{X}_\psi$. The simplified formulation for the real component of $\mathcal{F}_{DTCWT}(\cdot)$ is:

$$\mathbf{X}_F(u,v) = \mathbf{X}_\phi(u,v) + \mathbf{X}_\psi(u,v) = \sum_{h=0}^{H-1}\sum_{w=0}^{W-1} c_{M,h,w}\phi_{M,h,w} + \sum_{m=0}^{M-1}\sum_{h=0}^{H-1}\sum_{w=0}^{W-1}\sum_{k=1}^{6} d_{m,h,w}^{k}\psi_{m,h,w}^{k} \quad (3)$$

$M$ refers to resolution/level of decomposition and $k$ refers to directional selectivity. Similarly, we compute transformation for the imaginary component of $\mathcal{F}_{DTCWT}(\cdot)$.

**B. Spectral Gating Network:**

We introduce a novel method, Spectral Gating Network (SGN), to extract spectral features from both low and high-frequency components of the scattering transform. Figure-1 shows the architecture of our method. We use learnable weight parameters to blend each frequency component, but we use different blending methods for low and high frequencies. For the low-frequency component $\mathbf{X}_\phi \in \mathcal{R}^{C \times H \times W}$, we use the Tensor Blending Method (TBM), which is a new technique. TBM blends $X_\phi$ with $W_\phi$ using elementwise tensor multiplication, also known as Hadamard tensor product.

$$\mathcal{M}_\phi = [\mathbf{X}_\phi \odot \mathcal{W}_\phi], \qquad \text{where } (\mathbf{X}_\phi, \mathcal{W}_\phi) \in \mathcal{R}^{C \times H \times W}, \text{ and } \mathbf{M}_\phi \in \mathcal{R}^{C \times H \times W}, \quad (4)$$

$\mathcal{W}_\phi$ having same dimension as in $\mathcal{X}_\phi$. $\mathcal{M}_\phi$ is the low-frequency representation of the image and it captures global information of the image. One of the biggest challenges to getting effective features in the high-frequency components $\mathbf{X}_\psi \in \mathcal{R}^{k \times C \times H \times W \times 2}$, which are complex-valued and have 'k' times more dimensions than the low-frequency components ($\mathbf{X}_\phi$). Therefore, using the same Tensor Blending Method for the high-frequency components $X_\psi$ would increase the number of parameters by 2k times and also the computational cost (GFLOPS), where $k$ refers to directional selectivity, a factor of '2' indicating complex value comprising real and imaginary. To address this issue, we propose a new technique, the Einstein Blending Method (EBM), to blend the high-frequency components $X_\psi$ with the learnable weight parameters $W_\psi$ efficiently and effectively in the Spectral Gating Network that we propose in this paper. By using EBM, we can capture the fine-grain information in the image, such as texture, patterns, and small features.

To perform EBM, we first reshape a tensor $\mathbf{A}$ from $\mathbb{R}^{H \times W \times C}$ to $\mathbb{R}^{H \times W \times C_b \times C_d}$, where $C = C_b \times C_d$, and $b >> d$. We then define a weight matrix of size $W \in \mathbb{R}^{C_b \times C_d \times C_d}$. We then perform Einstein multiplication between $\mathbf{A}$ and $W$ along the last two dimensions, resulting in a blended feature tensor $Y \in \mathbb{R}^{H \times W \times C_b \times C_d}$ as shown in the Figure-2. The formula for EBM is:

$$\mathbf{Y}^{H \times W \times C_b \times C_d} = \mathbf{A}^{H \times W \times C_b \times C_d} \boxtimes \mathbf{W}^{C_b \times C_d \times C_d}$$

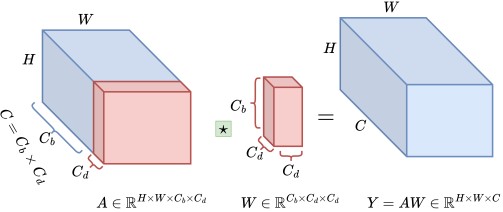

$$A \in \mathbb{R}^{H \times W \times C_b \times C_d} \qquad W \in \mathbb{R}^{C_b \times C_d \times C_d} \qquad Y = AW \in \mathbb{R}^{H \times W \times C}$$

Figure 2: Einstein Blending Method

## C. Spectral Channel and Token Mixing:

We perform EBM in the channel dimension of the high-frequency component we call Spectral Channel Mixing and following that we perform EBM in the token dimension of the high-frequency component, which we call Spectral Token Mixing. To perform EBM in the channel dimension, we first reshape the high-frequency component $X_\psi$ from $\mathbb{R}^{2 \times k \times H \times W \times C}$ to $\mathbb{R}^{2 \times k \times H \times W \times C_b \times C_d}$, where $C = C_b \times C_d$, and $b >> d$. We then define a weight matrix of size $W_{\psi_c} \in \mathbb{R}^{C_b \times C_d \times C_d}$. We then perform Einstein multiplication between $X_\psi$ and $W_{\psi_c}$ along the last two dimensions, resulting in a blended feature tensor $\mathbf{S}_{\psi_c} \in \mathbb{R}^{2 \times k \times H \times W \times C_b \times C_d}$. The formula for EBM in Channel mixing is:

$$\mathbf{S}_{\psi_c}^{2 \times k \times H \times W \times C_b \times C_d} = \mathbf{X}_\psi^{2 \times k \times H \times W \times C_b \times C_d} \boxtimes \mathbf{W}_{\psi_\mathbf{c}}^{C_b \times C_d \times C_d} + b_{\psi_c} \qquad (5)$$

To perform EBM in the Token dimension, we first reshape the high-frequency component $S_{\psi_c}$ from $\mathbb{R}^{2 \times k \times H \times W \times C}$ to $\mathbb{R}^{2 \times k \times C \times W \times H}$, where $Height(H) = Width(W)$. We then define a weight matrix of size $W_{\psi_t} \in \mathbb{R}^{W \times H \times H}$. We then perform Einstein multiplication between $X_\psi$ and $W_{\psi_t}$ along the last two dimensions, resulting in a blended feature tensor $\mathbf{S}_{\psi_t} \in \mathbb{R}^{2 \times k \times C \times W \times H}$. The formula for EBM in Token mixing is:

$$\mathbf{S}_{\psi_t}^{2 \times k \times C \times W \times H} = \mathbf{S}_{\psi_c}^{2 \times k \times C \times W \times H} \boxtimes \mathbf{W}_{\psi_\mathbf{t}}^{W \times H \times H} + b_{\psi_t} \qquad (6)$$

Where $\boxtimes$ represents an Einstein multiplication, the bias terms $b_{\psi_c} \in \mathbb{R}^{C_b \times C_d}$, $b_{\psi_t} \in \mathbb{R}^{H \times H}$. Now the total number of weight parameters in the high-frequency gating network is $(C_b \times C_d \times C_d) + (W \times H \times H)$ instead of $(C \times H \times W \times k \times 2)$ where $C >> H$ and bias is $(C_b \times C_d) + (H \times W)$. This reduces the number of parameters and multiplication while performing high-frequency gating operations in an image. We use a standard torch package [48] to perform Einstin multiplication. Finally, we perform inverse scattering transform using low-frequency representation( 4) and high-frequency representation( 6) to bring back the spectral domain to the physical domain. Our SVT architecture consists of L layers, comprising $\alpha$ scatter layers and $(L - \alpha)$ attention layers [59], where L denotes the network's depth. The scatter layers, being invertible, adeptly capture both the global and the fine-grain information in the image effectively via low-pass and high-pass filters, while attention layers focus on extracting semantic features and addressing long-range dependencies present in the image.

## 3 Experiment and Performance Studies

We evaluated SVT through various mainstream computer vision tasks including image recognition, object detection, and instance segmentation. To compare the quality of SVT transformer features, we conducted the following evaluations on standard datasets: a) We trained and evaluated ImageNet1K [10] from scratch for image recognition task. b) We performed transfer learning on CIFAR-10 [30], CIFAR-100 [30], Stanford Car [29], and Oxford Flower-102 [41] for Image recognition task. c) We conducted ablation studies to analyze variants of SVT transformers using scatter net with the help of various spectral mixing techniques. We also compare our results with transformers having similar decomposition architecture. d) We fine-tune SVT for downstream instance segmentation tasks. e) We also perform an in-depth analysis of the SVT, by conducting layer-wise analysis as well as invertibility analysis as well as latency analysis, and comparison.

### 3.1 Comparison with Similar architectures

We compare SVT with LiTv2 (Hilo) [42] which decomposes attention to find low and high-frequency components. We show that LiTv2 has a top-1 accuracy of 83.3%, while SVT has a top-1 accuracy of

Table 1: The table shows the performance of various vision backbones on the ImageNet1K[10] dataset for image recognition tasks. ★ indicates additionally trained with the Token Labeling objective using MixToken[25] and a convolutional stem (conv-stem) [60] for patch encoding. This table provides results for input image size $224 \times 224$. We have grouped the vision models into three categories based on their GFLOPs (Small, Base, and Large). The GFLOP ranges: Small (GFLOPs<6), Base (6≤GFLOPs<10), and Large (10≤GFLOPs<30).

| Method | Params | GFLOPS | Top-1 | Top-5 | Method | Params | GFLOPS | Top-1 | Top-5 |
|---|---|---|---|---|---|---|---|---|---|
| Small | | | | | Large | | | | |
| ResNet-50 [21] | 25.5M | 4.1 | 78.3 | 94.3 | ResNet-152 [21] | 60.2M | 11.6 | 81.3 | 95.5 |
| BoTNet-S1-50 [52] | 20.8M | 4.3 | 80.4 | 95.0 | ResNeXt101 [67] | 83.5M | 15.6 | 81.5 | - |
| Cross-ViT-S [6] | 26.7M | 5.6 | 81.0 | - | gMLP-B [37] | 73.0M | 15.8 | 81.6 | - |
| Swin-T [39] | 29.0M | 4.5 | 81.2 | 95.5 | DeiT-B [56] | 86.6M | 17.6 | 81.8 | 95.6 |
| ConViT-S [14] | 27.8M | 5.4 | 81.3 | 95.7 | SE-ResNet-152 [23] | 66.8M | 11.6 | 82.2 | 95.9 |
| T2T-ViT-14 [72] | 21.5M | 4.8 | 81.5 | 95.7 | Cross-ViT-B [6] | 104.7M | 21.2 | 82.2 | - |
| RegionViT-Ti+ [5] | 14.3M | 2.7 | 81.5 | - | ResNeSt-101 [75] | 48.3M | 10.2 | 82.3 | - |
| SE-CoTNetD-50 [35] | 23.1M | 4.1 | 81.6 | 95.8 | ConViT-B [14] | 86.5M | 16.8 | 82.4 | 95.9 |
| Twins-SVT-S [9] | 24.1M | 2.9 | 81.7 | 95.6 | PoolFormer [71] | 73.0M | 11.8 | 82.5 | - |
| CoaT-Lite-S [68] | 20.0M | 4.0 | 81.9 | 95.5 | T2T-ViTt-24 [72] | 64.1M | 15.0 | 82.6 | 95.9 |
| PVTv2-B2 [62] | 25.4M | 4.0 | 82.0 | 96.0 | TNT-B [19] | 65.6M | 14.1 | 82.9 | 96.3 |
| LITv2-S [42] | 28.0M | 3.7 | 82.0 | - | CycleMLP-B4 [7] | 52.0M | 10.1 | 83.0 | - |
| MViTv2-T [33] | 24.0M | 4.7 | 82.3 | - | DeepViT-L [78] | 58.9M | 12.8 | 83.1 | - |
| Wave-ViT-S [70] | 19.8M | 4.3 | 82.7 | 96.2 | RegionViT-B [5] | 72.7M | 13.0 | 83.2 | 96.1 |
| CSwin-T [12] | 23.0M | 4.3 | 82.7 | - | CycleMLP-B5 [7] | 76.0M | 12.3 | 83.2 | - |
| DaViT-Ti [11] | 28.3M | 4.5 | 82.8 | - | ViP-Large/7 [22] | 88.0M | 24.4 | 83.2 | - |
| SVT-H-S | 21.7M | 3.9 | 83.1 | 96.3 | CaiT-S36 [57] | 68.4M | 13.9 | 83.3 | - |
| iFormer-S [51] | 20.0M | 4.8 | 83.4 | 96.6 | AS-MLP-B [36] | 88.0M | 15.2 | 83.3 | - |
| CMT-S [17] | 25.1M | 4.0 | 83.5 | - | BoTNet-S1-128 [52] | 75.1M | 19.3 | 83.5 | 96.5 |
| MaxViT-T [58] | 31.0M | 5.6 | 83.6 | - | Swin-B [39] | 88.0M | 15.4 | 83.5 | 96.5 |
| Wave-ViT-S★ [70] | 22.7M | 4.7 | 83.9 | 96.6 | Wave-MLP-B [53] | 63.0M | 10.2 | 83.6 | - |
| **SVT-H-S★ (Ours)** | **22.0M** | **3.9** | **84.2** | **96.9** | LITv2-B [42] | 87.0M | 13.2 | 83.6 | - |
| Base | | | | | PVTv2-B4 [62] | 62.6M | 10.1 | 83.6 | 96.7 |
| ResNet-101 [21] | 44.6M | 7.9 | 80.0 | 95.0 | ViL-Base [76] | 55.7M | 13.4 | 83.7 | - |
| BoTNet-S1-59 [52] | 33.5M | 7.3 | 81.7 | 95.8 | Twins-SVT-L [9] | 99.3M | 15.1 | 83.7 | 96.5 |
| T2T-ViT-19 [72] | 39.2M | 8.5 | 81.9 | 95.7 | Hire-MLP-L [18] | 96.0M | 13.4 | 83.8 | - |
| CvT-21 [64] | 32.0M | 7.1 | 82.5 | - | RegionViT-B+ [5] | 73.8M | 13.6 | 83.8 | - |
| GFNet-H-B [47] | 54.0M | 8.6 | 82.9 | 96.2 | Focal-Base [69] | 89.8M | 16.0 | 83.8 | 96.5 |
| Swin-S [39] | 50.0M | 8.7 | 83.2 | 96.2 | PVTv2-B5 [62] | 82.0M | 11.8 | 83.8 | 96.6 |
| Twins-SVT-B [9] | 56.1M | 8.6 | 83.2 | 96.3 | CoTNetD-152 [35] | 55.8M | 17.0 | 84.0 | 97.0 |
| CoTNetD-101 [35] | 40.9M | 8.5 | 83.2 | 96.5 | DAT-B [65] | 88.0M | 15.8 | 84.0 | - |
| PVTv2-B3 [62] | 45.2M | 6.9 | 83.2 | 96.5 | LV-ViT-M★ [25] | 55.8M | 16.0 | 84.1 | 96.7 |
| LITv2-M [42] | 49.0M | 7.5 | 83.3 | - | CSwin-B [12] | 78.0M | 15.0 | 84.2 | - |
| RegionViT-M+ [5] | 42.0M | 7.9 | 83.4 | - | HorNet-$B_{GF}$ [46] | 88.0M | 15.5 | 84.3 | - |
| MViTv2-S [33] | 35.0M | 7.0 | 83.6 | - | DynaMixer-L [63] | 97.0M | 27.4 | 84.3 | - |
| CSwin-S [12] | 35.0M | 6.9 | 83.6 | - | MViTv2-B [33] | 52.0M | 10.2 | 84.4 | - |
| DaViT-S [11] | 49.7M | 8.8 | 84.2 | - | DaViT-B [11] | 87.9M | 15.5 | 84.6 | - |
| VOLO-D1★ [73] | 26.6M | 6.8 | 84.2 | - | CMT-L [17] | 74.7M | 19.5 | 84.8 | - |
| CMT-B [17] | 45.7M | 9.3 | 84.5 | - | MaxViT-B [58] | 120.0M | 23.4 | 85.0 | - |
| MaxViT-S [58] | 69.0M | 11.7 | 84.5 | - | VOLO-D2★ [73] | 58.7M | 14.1 | 85.2 | - |
| iFormer-B [51] | 48.0M | 9.4 | 84.6 | 97.0 | VOLO-D3★ [73] | 86.3M | 20.6 | 85.4 | - |
| Wave-ViT-B★ [70] | 33.5M | 7.2 | 84.8 | 97.1 | Wave-ViT-L★ [70] | 57.5M | 14.8 | 85.5 | 97.3 |
| **SVT-H-B★ (Ours)** | **32.8M** | **6.3** | **85.2** | **97.3** | **SVT-H-L★ (Ours)** | **54.0M** | **12.7** | **85.7** | **97.5** |

85.2% with a fewer number of parameters. We also compare SVT with iFormer [51] which captures low and high-frequency information from visual data, whereas SVT uses an invertible spectral method, namely the scattering network, to get the low-frequency and high-frequency components and uses tensor and Einstein mixing respectively to capture effective spectral features from visual data. SVT top-1 accuracy is 85.2, which is better than iFormer-B, which is at 84.6 with a lesser number of parameters and FLOPS. We compare SVT with WaveMLP [53] which is an MLP mixer-based technique that uses amplitude and phase information to represent the semantic content of an image. SVT uses a low-frequency component as an amplitude of the original feature, while a high-frequency component captures complex semantic changes in the input image. Our studies have shown, as depicted in Table- 1, that SVT outperforms WaveMLP by about 1.8%.

## 3.2 Comparison with State of the art methods

We divide the transformer architecture into three parts based on the computation requirements (FLOP counts) - small (less than 6 GFLOPS), base (6-10 GFLOPS), and large (10-30 GFLOPS). We use a similar categorization as WaveViT [70]. Notable recent works falling into the small category include C-Swin Transformers [12], LiTv2[42], MaxVIT[58], iFormer[51], CMT transformer, PVTv2[62], and WaveViT[70]. It's worth mentioning that WaveViT relies on extra annotations to achieve its best results. In this context, SVT-H-S stands out as the state-of-the-art model in the small category,

Table 2: **Initial Attention Layer vs Scatter Layer vs Initial Convolutional:** This table compares SVT transformer where initial scatter layers and later attention layers, SVT-Inverse where initial attention layers and later scatter layers, and SVT with initial convolutional layers. Also, we show an alternative spectral layer and attention layer. This shows that the Initial scatter layer works better compared to the rest.

| Model | Params(M) | FLOPS(G) | Top-1(%) | Top-5(%) |
|---|---|---|---|---|
| SVT-H-S | 22.0M | 3.9 | 84.2 | 96.9 |
| SVT-H-S-Init-CNN | 21.7M | 4.1 | 84.0 | 95.7 |
| SVT-H-S-Inverse | 21.8M | 3.9 | 83.1 | 94.6 |
| SVT-H-S-Alternate | 22.4M | 4.6 | 83.4 | 95.0 |

Table 3: This table shows the ablation analysis of various spectral layers in SVT architecture such as FN, FFC, WGN, and FNO. We conduct this ablation study on the small-size networks in stage architecture. This indicates that SVT performs better than other kinds of networks.

| Model | Params (M) | FLOPS (G) | Top-1 (%) | Top-5 (%) | Invertible loss($\downarrow$) |
|---|---|---|---|---|---|
| FFC | 21.53 | 4.5 | 83.1 | 95.23 | – |
| FN | 21.17 | 3.9 | 84.02 | 96.77 | – |
| FNO | 21.33 | 3.9 | 84.09 | 96.86 | 3.27e-05 |
| WGN | 21.59 | 3.9 | 83.70 | 96.56 | 8.90e-05 |
| SVT | 22.22 | 3.9 | **84.20** | 96.93 | 6.64e-06 |

Table 4: **Results on transfer learning datasets.** We report the top-1 accuracy on the four datasets.

| Model | CIFAR 10 | CIFAR 100 | Flowers 102 | Cars 196 |
|---|---|---|---|---|
| ResNet50 [21] | - | - | 96.2 | 90.0 |
| ViT-B/16 [13] | 98.1 | 87.1 | 89.5 | - |
| ViT-L/16 [13] | 97.9 | 86.4 | 89.7 | - |
| Deit-B/16 [56] | 99.1 | 90.8 | 98.4 | 92.1 |
| ResMLP-24 [55] | 98.7 | 89.5 | 97.9 | 89.5 |
| GFNet-XS [47] | 98.6 | 89.1 | 98.1 | 92.8 |
| GFNet-H-B [47] | 99.0 | 90.3 | 98.8 | 93.2 |
| SVT-H-B | **99.22** | **91.2** | **98.9** | **93.6** |

Table 5: The performances of various vision backbones on COCO val2017 dataset for the downstream instance segmentation task such as Mask R-CNN 1x [20] method. We adopt Mask R-CNN as the base model, and the bounding box & mask Average Precision (*i.e.*, $AP^b$ & $AP^m$) are reported for evaluation

| Backbone | $AP^b$ | $AP^b_{50}$ | $AP^b_{75}$ | $AP^m$ | $AP^m_{50}$ | $AP^m_{75}$ |
|---|---|---|---|---|---|---|
| ResNet50 [21] | 38.0 | 58.6 | 41.4 | 34.4 | 55.1 | 36.7 |
| Swin-T [39] | 42.2 | 64.6 | 46.2 | 39.1 | 61.6 | 42.0 |
| Twins-SVT-S [9] | 43.4 | 66.0 | 47.3 | 40.3 | 63.2 | 43.4 |
| LITv2-S [42] | 44.9 | - | - | 40.8 | - | - |
| RegionViT-S [5] | 44.2 | - | - | 40.8 | - | - |
| PVTv2-B2 [62] | 45.3 | 67.1 | 49.6 | 41.2 | 64.2 | 44.4 |
| SVT-H-S | **46.0** | **68.1** | **50.4** | **41.9** | **65.0** | **45.1** |

achieving a top-1 accuracy of 84.2%. Similarly, SVT-H-B surpasses all the transformers in the base category, boasting a top-1 accuracy of 85.2%. Lastly, SVT-H-L outperforms other large transformers with a top-1 accuracy of 85.7% when tested on the ImageNet dataset with an image size of 224x224.

When comparing different architectural approaches, such as Convolutional Neural Networks (CNNs), Transformer architectures (attention-based models), MLP Mixers, and Spectral architectures, SVT consistently outperforms its counterparts. For instance, SVT achieves better top-1 accuracy and parameter efficiency compared to CNN architectures like ResNet 152 [21], ResNeXt [67], and ResNeSt in terms of top-1 accuracy and number of parameters. Among attention-based architectures, MaxViT [58] has been recognized as the best performer, surpassing models like DeiT [56], Cross-ViT [6], DeepViT [78], T2T [72] etc. with a top-1 accuracy of 85.0. However, SVT achieves an even higher top-1 accuracy of 85.7 with less than half the number of parameters. In the realm of MLP Mixer-based architectures, DynaMixer [63] emerges as the top-performing model, surpassing MLP-mixer[54], gMLP [37], CycleMLP [7], Hire-MLP[18], AS-MLP [36], WaveMLP[53], PoolFormer[71] and DynaMixer-L [63] with a top-1 accuracy of 84.3%. In comparison, SVT-H-L outperforms DynaMixer with a top-1 accuracy of 85.7% while requiring fewer parameters and computations. Hierarchical architectures, which include models like PVT [61], Swin [39] transformer, CSwin [12] transformer, Twin [9] transformer, and VOLO [73] are also considered. Among this category, VOLO achieves the highest top-1 accuracy of 85.4%. However, it's important to note that SVT outperforms VOLO with a top-1 accuracy of 85.7% for SVT-H-L. Lastly, in the spectral architecture category, models like GFNet[47], iFormer [51], LiTv2 [42], HorNet [46], Wave-ViT [70], etc. are examined. Wave-ViT was previously the state-of-the-art method with a top-1 accuracy of 85.5%. Nevertheless, SVT-H-L surpasses Wave-ViT in terms of top-1 accuracy, network size (number of parameters), and computational complexity (FLOPS), as indicated in Table 1.

## 3.3 What Matters: Does Initial Spectral or Initial Attention or Initial Convolution Layers?

The ablation study was conducted to show that initial scatter layers followed by attention in deeper layers are more beneficial than having later scatter and initial attention layers ( SVT-H-S-Inverse). We also compare transformer models based on an alternative to the attention and scatter layer (SVT-H-S-Alternate) as shown in Table- 2. From all these combinations we observe that initial scatter layers followed by attention in deeper layers are more beneficial than others. We compare the performance of SVT when the architecture changes from all attention (PVTv2[62] ) to all spectral layers (GFNet[47])

Table 6: SVT model comprises low-frequency component and High-frequency component with the help of scattering net using Dual tree complex wavelet transform. Each frequency component is controlled by a parameterized weight matrix using Patch mixing and/or Channel Mixing. this table shows details about all combinations and $SVT_{TTEE}$ is the best performing among them.

| Backbone | Low Frequency | | High Frequency | | Params (M) | FLOPS (G) | Top-1 (%) | Top-5 (%) |
|---|---|---|---|---|---|---|---|---|
| | Token | Channel | Token | Channel | | | | |
| $SVT_{TTTT}$ | T | T | T | T | 25.18 | 4.4 | 83.97 | 96.86 |
| $SVT_{EETT}$ | E | E | T | T | 21.90 | 4.1 | 83.87 | 96.67 |
| $SVT_{EEEE}$ | E | E | E | E | 21.87 | 3.7 | 83.70 | 96.56 |
| $SVT_{TTEE}$ | T | T | E | E | 22.01 | 3.9 | 84.20 | 96.82 |
| $SVT_{TTEX}$ | T | T | E | ✗ | 21.99 | 4.0 | 84.06 | 96.76 |
| $SVT_{TTXE}$ | T | T | ✗ | E | 22.25 | 4.1 | 84.12 | 96.91 |

as well as a few spectral and remaining attention layers(SVT ours). We observe that combining spectral and attention boost the performance compared to all attention and all spectral layer-based transformer as shown in Table- 2. We have conducted an experiment where the initial layers of a ViT are convolutional networks and later layers are attention layers to compare the performance of SVT. The results are captured in Table- 1, where we compare SVT with transformers having initial convolutional layers such as CVT [64], CMT [17], and HorNet [46]. Initial convolutional layers in a transformer are not performing well compared to the initial scatter layer. Initial scatter layer-based transformers have better performance and less computation cost compared to initial convolutional layer-based transformers which is shown in Table- 2.

## 3.4  Ablation analysis

SVT uses a scattering network to decompose the signal into low-frequency and high-frequency components. We use a gating operator to get effective learnable features for spectral decomposition. The gating operator is a multiplication of the weight parameter in both high and low frequencies. We have conducted experiments that use tensor and Einstein mixing. Tensor mixing is a simple multiplication operator, while Einstein mixing uses an Einstein matrix multiplication operator [48]. We observe that in low-frequency components, Tensor mixing performs better as compared to Einstein mixing. As shown in Table- 6, we start with $SVT_{TTTT}$, which uses tensor mixing in both high and low-frequency components. We see that it may not perform optimally. Then we reverse it and use Einstein mixing in both low and high-frequency components - this also does not perform optimally. Then, we came up with the alternative method $SVT_{TTEE}$, which uses tensor mixing in low frequency and Einstein mixing in high frequency. The high-frequency further decomposes into token and channel mixing, whereas in low-frequency we simply tensor multiplication as it is an energy or amplitude component.

In the second ablation analysis, we compare various spectral architectures, including the Fourier Network (FN), Fourier Neural Operator (FNO), Wavelet Gating Network (WGN), and Fast Fourier Convolution (FFC). When we contrast SVT with WGN, it becomes evident that SVT exhibits superior directional selectivity and a more adept ability to manage complex transformations. Furthermore, in comparison to FN and FNO, SVT excels in decomposing frequencies into low and high-frequency components. It's worth noting that SVT surpasses other spectral architectures primarily due to its utilization of the Directional Dual-Tree Complex Wavelet Transform (DTCWT), which offers directional orientation and enhanced invertibility, as demonstrated in Table 3. For a more comprehensive analysis, please refer to the Supplementary section.

## 3.5  Transfer Learning and Task Learning

We train SVT on ImageNet1K data and fine-tune it on various datasets such as CIFAR10, CIFAR100, Oxford Flower, and Stanford Car for image recognition tasks. We compare SVT-H-B performance with various transformers such as Deit [56], ViT [13], and GFNet [47] as well as with CNN architectures such as ResNet50 and MLP mixer architectures such as ResMLP. This comparison is shown in Table- 4. It can be observed that SVT-H-B outperforms state-of-art on CIFAR10 with a top-1 accuracy of 99.1%, CIFAR100 with a top-1 accuracy of 91.3%, Flowers with a top-1 accuracy of 98.9% and Cars with top-1 accuracy of 93.7%. We observe that SVT has more representative features and has an inbuilt discriminative nature which helps in classifying images into various categories. We use a

Table 7: **Latency(Speed test):** This table shows the Latency (mili sec) of SVT compared with Conv type network, attention type transformer, POOl type, MLP type, and Spectral type transformer. We report latency per sample on A100 GPU. We adopt the latency table from EfficientFormer [34].

| Model | Type | Params (M) | GMAC (G) | Top-1 (%) | Latency (ms) |
|---|---|---|---|---|---|
| ResNet50[21] | Convolution | 25.5 | 4.1 | 78.5 | 9.0 |
| DeiT-S[56] | Attention | 22.5 | 4.5 | 81.2 | 15.5 |
| PVT-S[62] | Attention | 24.5 | 3.8 | 79.8 | 23.8 |
| T2T-14[] | Attention | 21.5 | 4.8 | 81.5 | 21.0 |
| Swin-T[38] | Attention | 29.0 | 4.5 | 81.3 | 22.0 |
| CSwin-T[12] | Attention | 23.0 | 4.3 | 82.7 | 28.7 |
| PoolFormer[71] | Pool | 31.0 | 5.2 | 81.4 | 41.2 |
| ResMLP-S[55] | MLP | 30.0 | 6.0 | 79.4 | 17.4 |
| EfficientFormer [34] | MetaBlock | 31.3 | 3.9 | 82.4 | 13.9 |
| GFNet-H-S[47] | Spectral | 32.0 | 4.6 | 81.5 | 14.3 |
| SVT-H-S | Spectral | 22.0 | 3.9 | 84.2 | 14.7 |

Table 8: **Invertibility:** This table shows the invertibility of SVT(DTCWT) compared with Fourier and DWT. We also compare different directional orientations and show the reconstruction loss (MSE) in an image.

| Model | MSE loss($\downarrow$) | PSNR (db)($\uparrow$) |
|---|---|---|
| Fourier (FFT) | 3.27e-05 | 11.18 |
| DWT-M1 | 8.90e-05 | 76.33 |
| DWT-M2 | 3.19e-05 | 84.67 |
| DWT-M3 | 1.08e-05 | 91.94 |
| DTCWT-M1 | 6.64e-06 | 137.97 |
| DTCWT-M2 | 2.01e-06 | 138.87 |
| DTCWT-M3 | 1.23e-07 | 142.14 |

pre-trained SVT model for the downstream instance segmentation task and obtain good results on the MS-COCO dataset as shown in Table- 5.

## 3.6 Latency Analysis

It's important to highlight that Fourier Transforms, as mentioned in the GFNet[47], are not inherently capable of performing low-pass and high-pass separations. In contrast, GFNet consistently employs tensor multiplication, a method that, while effective, may be less efficient compared to Einstein multiplication. The latter approach is known for reducing the number of parameters and computational complexity. As a result, SVT does not lag behind in terms of performance or computational complexity; rather, it gains enhanced representational power. This is exemplified in Table 7, which provides a comparison of latency, FLOPS (Floating-Point Operations per Second), and the number of parameters. Table 7 specifically demonstrates the latency (measured in milliseconds) of SVT in relation to various network types, including convolution-based networks, Attention-based Transformer networks, Pool-based Transformer networks, MLP-based Transformer networks, and Spectral-based Transformer networks. The reported latency values are on a per-sample basis, measured on an A100 GPU.

## 3.7 Invertibility Versus Redundancy trade-off Analysis

We conducted an experiment to illustrate that invertibility not only enhances performance but also contributes to image comprehension. To do this, we passed an image through the raw DTC-WT and performed an inverse DTC-WT operation to calculate the reconstruction loss. The experiment was executed across various values of "M" corresponding to the level of decomposition and orientations in SVT. We observed that the reconstruction loss decreased as the value of "M" increased, indicating that SVT's ability to comprehend the image improved. These orientations effectively captured higher-order image properties, enhancing SVT's performance. We further compared different spectral transforms, including the Fourier Transform, Discrete Wavelet Transform (DWT), and DTC-WT. Our findings demonstrated that the reconstruction loss was lower for DTC-WT compared to other spectral transforms, as depicted in Table 8 below. In Table 8, we quantified the mean squared error (MSE) for FFT, DWT at stages 1, 2, and 3, and DTCWT at stages 1, 2, and 3. The MSE decreased as we increased the level of decomposition (M) and the degree of selectivity. DTCWT consistently exhibited lower MSE compared to DWT. Furthermore, the peak signal-to-noise ratio (PSNR) of DTCWT surpassed that of DWT and the Fourier Transform. PSNR gauges the quality of the reconstructed image, expressing the ratio between the maximum possible power of an image and the power of noise affecting its representation, measured in decibels (dB). A higher PSNR indicates superior image quality. A high-quality reconstructed image is characterized by low MSE and high PSNR values. For further details on redundancy, please consult the supplementary table. Additionally, we have visualized the filter coefficients for all six orientations in the supplementary materials.

## 3.8 Limitations

SVT currently uses six directional orientations to capture an image's fine-grained semantic information. It is possible to go for the second degree, which gives thirty-six orientations, while the third degree gives 216 orientations. The more orientations, the more semantic information could be captured, but

this leads to higher computational complexity. The decomposition parameter 'M' is currently set to 1 to get single low-pass and high-pass components. Higher values of 'M' give more components in both frequencies but lead to higher complexity.

## 4    Related Work

The Vision Transformer (ViT) [13] was the first transformer-based attempt to classify images into pre-defined categories and use NLP advances in vision. Following this, several transformer based approaches like DeiT[56], Tokens-to-token ViT [72], Transformer iN Transformer (TNT) [19], Cross-ViT [6], Class attention image Transformer(CaiT) [57] Uniformer [32], Beit. [3], SViT[45], RegionViT [5], MaxViT [58] etc. have all been proposed to improve the accuracy using multi-headed self-attention (MSA). PVT [61], SwinT [39], CSwin[12] and Twin [39] use hierarchical architecture to improve the performance of the vision transformer on various tasks. The complexity of MSA is O($n^2$). For high-resolution images, the complexity increases quadratically with token length. PoolFormer [71] is a method that uses a pooling operation over a small patch which has to obtain a down-sampled version of the image to reduce computational complexity. The main problem with PoolFormer is that it uses a MaxPooling operation which is not invertible. Another approach to reducing the complexity is the spectral transformers such as FNet [31], GFNet [47], AFNO [16], WaveMix [24], WaveViT [70], SpectFormer [44], FourierFormer [40], etc. FNet [31] does not use inverse Fourier transforms, leading to an invertibility issue. GFNet [47] solves this by using inverse Fourier transforms with a gating network. AFNO [16] uses the adaptive nature of a Fourier neural operator similar to GFNet. SpectFormer [44] introduces a novel transformer architecture that combines both spectral and attention networks for vision tasks. GFNet, SpectFormer, and AFNO do not have proper separation of low-frequency and high-frequency components and may struggle to handle the semantic content of images. In contrast, SVT has a clear separation of frequency components and uses directional orientations to capture semantic information. FourierIntegral [40] is similar to GFNet and may have similar issues in separating frequency components.

WaveMLP [53] is a recent effort that dynamically aggregates tokens as a wave function with two parts, amplitude and phase to capture the original features and semantic content of the images respectively. SVT uses a scattered network to provide low-frequency and high-frequency components. The high-frequency component has six or more directional orientations to capture semantic information in images. We use Einstein multiplication in token and channel mixing of high-frequency components leading to lower computational complexity and network size. In Wave-ViT [70], the author has discussed the quadratic complexity of the self-attention network using a wavelet transform to perform lossless down-sampling using wavelet transform over keys and values. However, WaveViT still has the same complexity as it uses attention instead of spectral layers. SVT uses the scatter network which is more invertible compared to WaveViT.

One of the challenges in MSA is its inability to characterize different frequencies in the input image. Hilo attention (LiTv2) [42] helps to find high-frequency and low-frequency components by using a novel variant of MSA. But it does not solve the complexity issue of MSA. Another parallel effort named Inception Transformer came up [51], which uses an Inception mixer to capture high and low-frequency information in visual data. iFormer still has the same complexity as it uses attention as the low-frequency mixer. SVT in comparison, uses a spectral neural operator to capture low and high frequency components using the DTCWT. This removes the O($n^2$) complexity as it uses spectral mixing instead of attention. iFormer [51] uses a non-invertible max pooling and convolutional layer to capture high-frequency components, whereas, in contrast, SVT's mixer is completely invertible. SVT uses a scatter network to get a better directional orientation to capture fine-grained information such as lines and edges, compared to Hilo attention and iFormer.

## 5    Conclusions and Future Research Directions

We have proposed SVT, which helps in separating low-frequency and high-frequency components of an image, while simultaneously reducing computational complexity by using Einstein multiplication-based technique for efficient channel and token mixing. SVT has been evaluated on standard benchmarks and shown to achieve state-of-the-art performance on standard benchmark datasets on both image classification tasks and instance segmentation tasks. It also achieves comparable performance on object detection tasks. We shall experiment with SVT in other domains such as speech and NLP as we believe that it offers significant value in these domains as well.

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
