# Scattering Vision Transformer: Spectral Mixing Matters-Supplementary

**Badri Narayana Patro**
Microsoft
badripatro@microsoft.com

Vijay Srinivas Agneeswaran
Microsoft
vagneeswaran@microsoft.com

## 1 Introduction

This document provides a comprehensive analysis of the vanilla transformer architecture and explores various versions The architecture comparisons are presented in Table-4, shedding light on the differences and capabilities of each version. The document also delves into the training configurations, encompassing transfer learning, task learning, and fine-tuning tasks. The dataset information utilized for transformer learning is presented in Table- 5, providing insights into dataset sizes, and relevance to different applications. Moving to the results section, we showcase the fine-tuned model outcomes, where models are initially trained on 224 x 224 images and subsequently fine-tuned on 384 x 384 images. The performance evaluation, as depicted in Table- 6, encompasses accuracy metrics, number of parameters(M), and Floating point operations(G). The detailed comparison of similar architectures is provided in Table- 3. Regarding the trade-off between invertibility and redundancy, we conducted an experiment to demonstrate that invertibility aids in comprehending the image rather than merely contributing to performance, as shown in Table- 2.

## 2 Filter visualisation analysis

SVT incorporates the scattering network utilizing the DTCWT for image decomposition into low and high-frequency components. Our primary focus is to analyze the low-frequency and high-frequency filter components to emphasize SVT's exceptional directional orientation capabilities. It is worth noting that unlike other spectral transformers, such as GFNet, SVT exhibits pronounced directional orientation. To gain insights into SVT's performance, we visualize the first four layers of the SVT transformer, particularly focusing on 24 filter coefficients out of the total 384. Moreover, our analysis includes the examination of six directional components; however, we present only the first two directional components, along with the low-pass filter components for the purpose of brevity. Through these visualizations, we aim to showcase how SVT adeptly captures lines and edges with diverse orientations, outperforming other spectral transformers. The findings from our visual analysis, illustrated in Figure 1, provide compelling evidence of SVT's superiority in handling directional information compared to other spectral transformers.

The visualization of each directional component allows us to observe its ability to capture lines and edges with diverse orientations, surpassing the performance of other spectral transformers. To support our findings, Figure 2 exhibits the visual representations of these filter components, providing clear evidence of SVT's superior orientation handling capabilities. This analysis serves to score the significance of SVT's architecture in effectively extracting and leveraging directional information, contributing to its enhanced performance in various computer vision and signal processing tasks.

37th Conference on Neural Information Processing Systems (NeurIPS 2023).

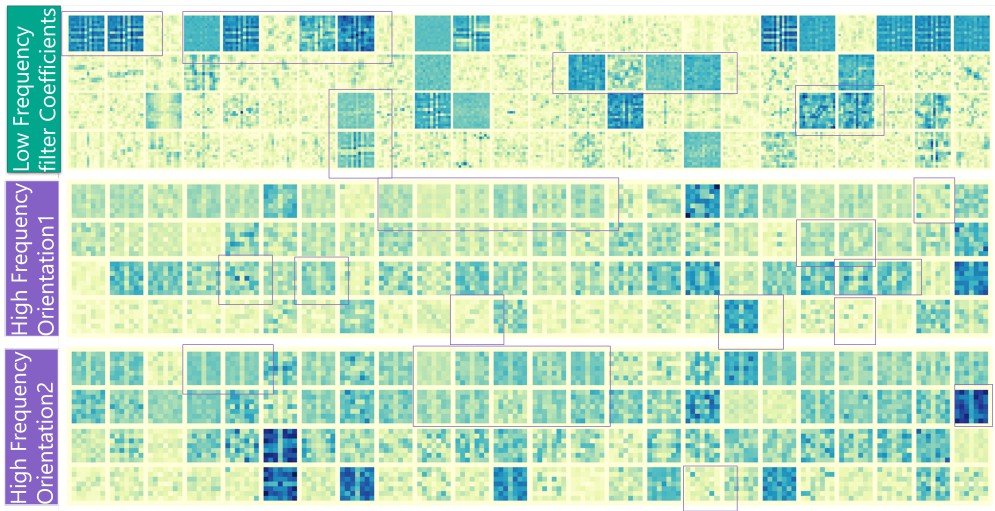

Figure 1: This figure shows the Filter characterization of the initial four layers of the SVT model. It clearly shows that High-frequency filter coefficient c captures local filter information such as lines, edges, and different orientations of an Image. The Low-frequency filter coefficient captures the shape with the maximum energy part in the image.

Table 1: Detailed architecture specifications for three variants of our SVT with different model sizes, *i.e.*, SVT-S (small size), SVT-B (base size), and SVT-L (large size). $E_i$, $G_i$, $H_i$, and $C_i$ represent the expansion ratio of the feed-forward layer, the spectral gating number, the head number, and the channel dimension in each stage $i$, respectively.

| | OP Size | SVT-H-S | SVT-H-B | SVT-H-L |
|---|---|---|---|---|
| Stage 1 | $\frac{H}{4} \times \frac{W}{4}$ | $\begin{bmatrix} E_1 = 8 \\ G_1 = 1 \\ C_1 = 64 \end{bmatrix} \times 3$ | $\begin{bmatrix} E_1 = 8 \\ G_1 = 1 \\ C_1 = 64 \end{bmatrix} \times 3$ | $\begin{bmatrix} E_1 = 8 \\ G_1 = 1 \\ C_1 = 96 \end{bmatrix} \times 3$ |
| Stage 2 | $\frac{H}{8} \times \frac{W}{8}$ | $\begin{bmatrix} E_2 = 8 \\ G_2 = 1 \\ C_2 = 128 \end{bmatrix} \times 4$ | $\begin{bmatrix} E_2 = 8 \\ G_2 = 1 \\ C_2 = 128 \end{bmatrix} \times 4$ | $\begin{bmatrix} E_2 = 8 \\ G_2 = 1 \\ C_2 = 192 \end{bmatrix} \times 6$ |
| Stage 3 | $\frac{H}{16} \times \frac{W}{16}$ | $\begin{bmatrix} E_3 = 4 \\ H_3 = 10 \\ C_3 = 320 \end{bmatrix} \times 6$ | $\begin{bmatrix} E_3 = 4 \\ H_3 = 10 \\ C_3 = 320 \end{bmatrix} \times 12$ | $\begin{bmatrix} E_3 = 4 \\ H_3 = 12 \\ C_3 = 384 \end{bmatrix} \times 18$ |
| Stage 4 | $\frac{H}{32} \times \frac{W}{32}$ | $\begin{bmatrix} E_4 = 4 \\ H_4 = 14 \\ C_4 = 448 \end{bmatrix} \times 3$ | $\begin{bmatrix} E_4 = 4 \\ H_4 = 16 \\ C_4 = 512 \end{bmatrix} \times 3$ | $\begin{bmatrix} E_4 = 4 \\ H_4 = 16 \\ C_4 = 512 \end{bmatrix} \times 3$ |

# 3  Dataset and Training Details:

## 3.1  Dataset and Training Setups on ImageNet-1K for Image Classification task

In this section, we outline the dataset and training setups for the Image Classification task on the ImageNet-1K benchmark dataset. The dataset comprises 1.28 million training images and 50K validation images, spanning across 1,000 categories. To train the vision backbones from scratch, we employ several data augmentation techniques, including RandAug, CutOut, and Token Labeling objectives

Table 2: **Invertibility vs redundancy:** This table shows the SVT-H performance for each orientation. We merge all the orientations and make them similar, making 2 and 3 orientations. Final SVT-H-S has 6 orientations in high-frequency components to capture curves and slants in all 6 orientations. 'H' stands for hierarchical, 'S' for small size mode for image size $224^2$

| Model | Params | GFLOPs | Top-1(%) | Top-5(%) |
|---|---|---|---|---|
| SVT-H-S-ori-1 | 21.5M | 3.9 | 83.2 | 94.9 |
| SVT-H-S-ori-2 | 21.6M | 3.9 | 83.4 | 95.1 |
| SVT-H-S-ori-3 | 21.7M | 3.9 | 83.7 | 95.5 |
| SVT-H-S(ori-6) | 22.0M | 3.9 | 84.2 | 96.9 |

Low-Frequency Filters coefficients

High-Frequency Filters coefficients-orientation-0

High-Frequency Filters coefficients-orientation-1

High-Frequency Filters coefficients-orientation-2

High-Frequency Filters coefficients-orientation-3

High-Frequency Filters coefficients-orientation-4

High-Frequency Filters coefficients-orientation-5

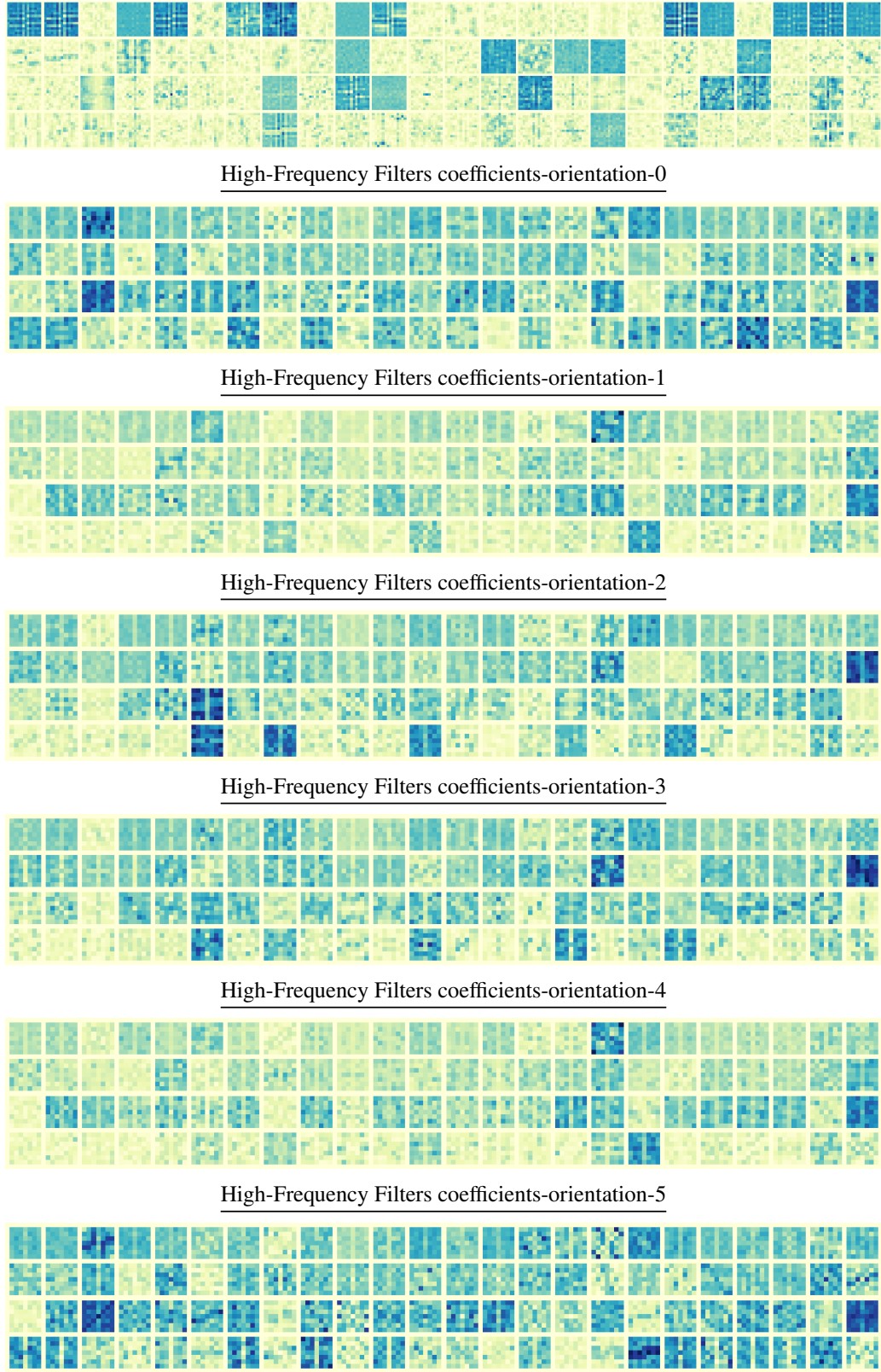

Figure 2: This figure shows the Filter characterization of the initial four layers of the SVT model. It clearly shows that the High-frequency filter coefficient captures local filter information such as lines, edges, and different orientations of an Image. The Low-frequency filter coefficient captures the shape with the maximum energy part in the image.

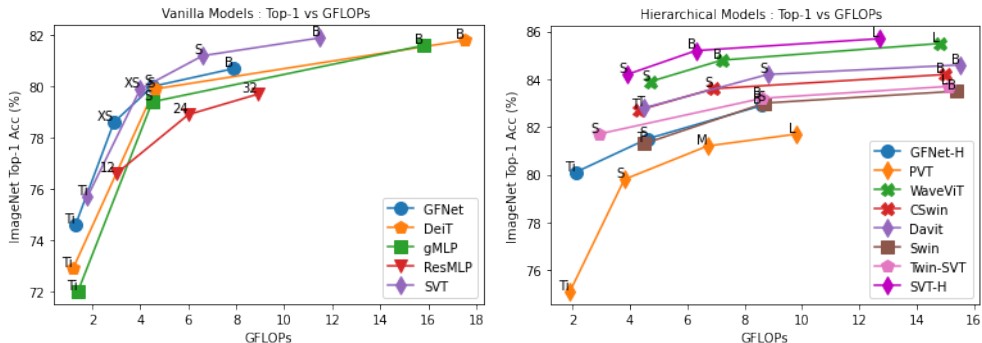

Figure 3: Comparison of ImageNet Top-1 Accuracy (%) vs GFLOPs of various models in Vanilla and Hierarchical architecture.

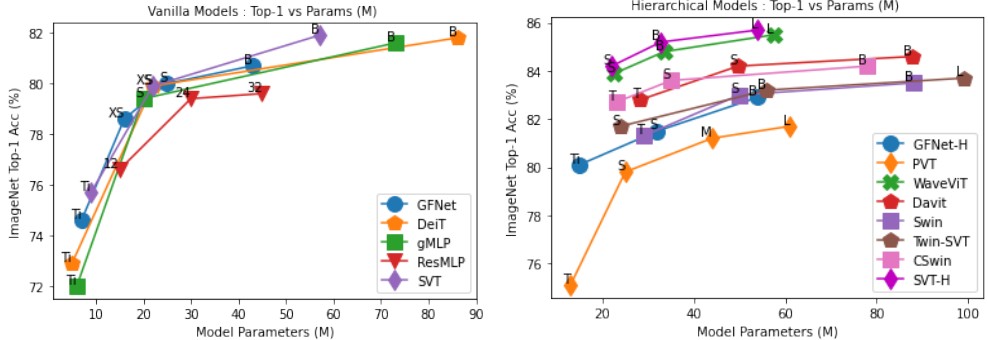

Figure 4: Comparison of ImageNet Top-1 Accuracy (%) vs Parameters (M) of various models in Vanilla and Hierarchical architecture.

with MixToken. These augmentation techniques help enhance the model's generalization capabilities. For performance evaluation, we measure the trained backbones' top-1 and top-5 accuracies on the validation set, providing a comprehensive assessment of the model's classification capabilities. In the optimization process, we adopt the AdamW optimizer with a momentum of 0.9, combining it with a 10-epoch linear warm-up phase and a subsequent 310-epoch cosine decay learning rate scheduler. These strategies aid in achieving stable and effective model training. To handle the computational load, we distribute the training process on 8 V100 GPUs, utilizing a batch size of 128. This distributed setup helps accelerate the training process while making efficient use of available hardware resources. The learning rate and weight decay are fixed at 0.00001 and 0.05, respectively, maintaining stable training and mitigating overfitting risks.

## 3.2 Training setup for Transfer Learning

In the context of transfer learning, we sought to evaluate the efficacy of our vanilla SVT architecture on widely-used benchmark datasets, namely CIFAR-10 [8], CIFAR100 [8], Oxford-IIIT-Flower [14] and Standford Cars [7]. Our approach followed the methodology of previous studies [20, 3, 23, 22, 18], where we initialized the model with pre-trained weights from ImageNet and subsequently fine-tuned it on the new datasets.

Table-4 in the main paper presents a comprehensive comparison of the transfer learning performance of both our basic and best models against state-of-the-art CNNs and vision transformers. To maintain consistency, we employed a batch size of 64, a learning rate (lr) of 0.0001, a weight-decay of 1e-4, a clip-grad value of 1, and performed 5 epochs of warmup. For the transfer learning process, we utilized a pre-trained model that was initially trained on the ImageNet-1K dataset. This pre-trained model was fine-tuned on the specific transfer learning dataset mentioned in Table-5 for a total of 1000 epochs.

Table 3: This shows a performance comparison of SVT with similar Transformer Architecture with different sizes of the networks on ImageNet-1K. ⋆ indicates additionally trained with the Token Labeling objective using MixToken[6].

| Network | Params | GFLOPs | Top-1 Acc (%) | Top-5 Acc (%) |
|---|---|---|---|---|
| Vanilla Transformer Comparison | | | | |
| FFC-ResNet-50 [1] | 26.7M | - | 77.8 | - |
| FourierFormer [13] | - | - | 73.3 | 91.7 |
| GFNet-Ti [18] | 7M | 1.3 | 74.6 | 92.2 |
| SVT-T | 9M | 1.8 | **76.9** | **93.4** |
| FFC-ResNet-101 [1] | 46.1M | - | 78.8 | - |
| Fnet-S [9] | 15M | 2.9 | 71.2 | - |
| GFNet-XS [18] | 16M | 2.9 | 78.6 | 94.2 |
| GFNet-S [18] | 25M | 4.5 | 80.0 | 94.9 |
| SVT-XS | 19.9M | 4.0 | **79.9** | **94.5** |
| SVT-S | 32.2M | 6.6 | **81.5** | **95.3** |
| FFC-ResNet-152 [1] | 62.6M | - | 78.9 | - |
| GFNet-B [18] | 43M | 7.9 | 80.7 | 95.1 |
| SVT-B | 57.6M | 11.8 | **82.0** | **95.6** |
| Hierarchical Transformer Comparison | | | | |
| GFNet-H-S [18] | 32M | 4.6 | 81.5 | 95.6 |
| LIT-S [16] | 27M | 4.1 | 81.5 | - |
| iFormer-S[19] | 20 | 4.8 | 83.4 | 96.6 |
| Wave-ViT-S⋆ [25] | 22.7M | 4.7 | 83.9 | 96.6 |
| SVT-H-S | 21.7M | 3.9 | 83.1 | 96.3 |
| SVT-H-S⋆ | 22.0M | 3.9 | **84.2** | **96.9** |
| GFNet-H-B [18] | 54M | 8.6 | 82.9 | 96.2 |
| LIT-M [16] | 48M | 8.6 | 83.0 | - |
| LITv2-M [15] | 49.0M | 7.5 | 83.3 | - |
| iFormer-B[19] | 48 | 9.4 | 84.6 | 97.0 |
| Wave-MLP-B [21] | 63.0M | 10.2 | 83.6 | - |
| Wave-ViT-B⋆ [25] | 33.5M | 7.2 | 84.8 | **97.0** |
| SVT-H-B⋆ | 32.8M | 6.3 | **85.2** | **97.3** |
| LIT-B [16] | 86M | 15.0 | 83.4 | - |
| LITv2-B [15] | 87.0M | 13.2 | 83.6 | - |
| HorNet-$B_{GF}$ [17] | 88.0M | 15.5 | 84.3 | - |
| iFormer-L[19] | 87.0M | 14.0 | 84.8 | **97.0** |
| Wave-ViT-L⋆ [25] | 57.5M | 14.8 | 85.5 | 97.3 |
| SVT-H-L⋆ | 54.0M | 12.7 | **85.7** | **97.5** |

## 3.3 Training setup for Task Learning

In this section, we conduct an in-depth analysis of the pre-trained SVT-H-small model's performance on the COCO dataset for two distinct downstream tasks involving object localization, ranging from bounding-box level to pixel level. Specifically, we evaluate our SVT-H-small model on instance segmentation tasks, such as Mask R-CNN [5], as demonstrated in Table-5 of the main paper.

For the downstream task, we replace the CNN backbones in the respective detectors with our pre-trained SVT-H-small model to evaluate its effectiveness. Prior to this, we pre-train each vision backbone on the ImageNet-1K dataset, initializing the newly added layers with Xavier initialization [4]. Next, we adhere to the standard setups defined in [11] to train all models on the COCO train2017 dataset, which comprises approximately 118,000 images. The training process is performed with a batch size of 16, and we utilize the AdamW optimizer [12] with a weight decay of 0.05, an initial learning rate of 0.0001, and betas set to (0.9, 0.999). To manage the learning rate during training, we adopt the step learning rate policy with linear warm-up at every 500 iterations and a warm-up

Table 4: In this table, we present a comprehensive overview of different versions of SVT within the vanilla transformer architecture. The table includes detailed configurations such as the number of heads, embedding dimensions, the number of layers, and the training resolution for each variant. For SVT-H models with a hierarchical structure, we refer readers to Table-4 in the main paper, which outlines the specifications for all four stages. Additionally, the table provides FLOPs (floating-point operations) calculations for input sizes of both 224×224 and 384×384. In the vanilla SVT architecture, we utilize four spectral layers with $\alpha = 4$, while the remaining attention layers are $(L - \alpha)$.

| Model | #Layers | #heads | #Embedding Dim | Params (M) | Training Resolution | FLOPs (G) |
|-------|---------|--------|----------------|------------|---------------------|-----------|
| SVT-Ti | 12 | 4 | 256 | 9 | 224 | 1.8 |
| SVT-XS | 12 | 6 | 384 | 20 | 224 | 4.0 |
| SVT-S | 19 | 6 | 384 | 32 | 224 | 6.6 |
| SVT-B | 19 | 8 | 512 | 57 | 224 | 11.5 |
| SVT-XS | 12 | 6 | 384 | 21 | 384 | 13.1 |
| SVT-S | 19 | 6 | 384 | 33 | 384 | 22.0 |
| SVT-B | 19 | 8 | 512 | 57 | 384 | 37.3 |

Table 5: This table presents information about datasets used for transfer learning. It includes the size of the training and test sets, as well as the number of categories included in each dataset.

| Dataset | CIFAR-10 [8] | CIFAR-100 [8] | Flowers-102 [14] | Stanford Cars [7] |
|---------|--------------|---------------|------------------|-------------------|
| Train Size | 50,000 | 50,000 | 8,144 | 2,040 |
| Test Size | 10,000 | 10,000 | 8,041 | 6,149 |
| #Categories | 10 | 100 | 196 | 102 |

ratio of 0.001. These learning rate configurations aid in optimizing the model's performance and convergence.

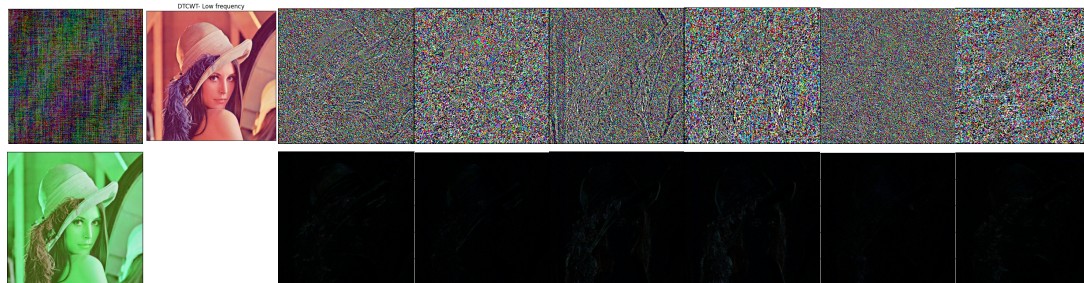

Figure 5: The 1st column shows phase and magnitude plots for the Fourier transformer and the 2nd column shows the low-frequency component of Dual tree Complex Wavelet transform (DT-CWT). 3rd column onwards shows high-frequency visualization of all 6 direction-selective. 1st row visualizes phase information & the second row shows the magnitude of all 6 high-frequency components.

## 3.4 Training setup for Fine-tuning task

In our main experiments, we conduct image classification tasks on the widely-used ImageNet dataset [2], a standard benchmark for large-scale image classification. To ensure a fair and meaningful comparison with previous research [23, 22, 18], we adopt the same training details for our SVT models. For the vanilla transformer architecture (SVT), we utilize the hyperparameters recommended by the GFNet implementation [18]. Similarly, for the hierarchical architecture (SVT-H), we employ the hyperparameters recommended by the WaveViT implementation [25]. During fine-tuning at higher resolutions, we follow the hyperparameters suggested by the GFNet implementation [18] and train our models for 30 epochs.

All model training is performed on a single machine equipped with 8 V100 GPUs. In our experiments, we specifically compare the fine-tuning performance of our models with GFNet [18]. Our observations

Table 6: We conducted a comparison of various transformer-style architectures for image classification on ImageNet. This includes **vision transformers [23], MLP-like models [22, 10], spectral transformers [18] and our SVT models**, which have similar numbers of parameters and FLOPs. The top-1 accuracy on ImageNet's validation set, as well as the number of parameters and FLOPs, are reported. All models were trained using $224 \times 224$ images. We used the notation "↑384" to indicate models fine-tuned on $384 \times 384$ images for 30 epochs.

| Model | Params (M) | FLOPs (G) | Resolution | Top-1 Acc. (%) | Top-5 Acc. (%) |
|---|---|---|---|---|---|
| gMLP-Ti [10] | 6 | 1.4 | 224 | 72.0 | - |
| DeiT-Ti [23] | 5 | 1.2 | 224 | 72.2 | 91.1 |
| GFNet-Ti [18] | 7 | 1.3 | 224 | 74.6 | 92.2 |
| SVT-T | 9 | 1.8 | 224 | 76.9 | 93.4 |
| ResMLP-12 [22] | 15 | 3.0 | 224 | 76.6 | - |
| GFNet-XS [18] | 16 | 2.9 | 224 | 78.6 | 94.2 |
| SVT-XS | 20 | 4.0 | 224 | 79.9 | 94.5 |
| DeiT-S [23] | 22 | 4.6 | 224 | 79.8 | 95.0 |
| gMLP-S [10] | 20 | 4.5 | 224 | 79.4 | - |
| GFNet-S [18] | 25 | 4.5 | 224 | 80.0 | 94.9 |
| SVT-S | 32 | 6.6 | 224 | 81.5 | 95.3 |
| ResMLP-36 [22] | 45 | 8.9 | 224 | 79.7 | - |
| GFNet-B [18] | 43 | 7.9 | 224 | 80.7 | 95.1 |
| gMLP-B [10] | 73 | 15.8 | 224 | 81.6 | - |
| DeiT-B [23] | 86 | 17.5 | 224 | 81.8 | 95.6 |
| SVT-B | 57 | 11.6 | 224 | **82.0** | **95.6** |
| GFNet-XS↑384 [18] | 18 | 8.4 | 384 | 80.6 | 95.4 |
| GFNet-S↑384 [18] | 28 | 13.2 | 384 | 81.7 | 95.8 |
| GFNet-B↑384 [18] | 47 | 23.3 | 384 | 82.1 | 95.8 |
| SVT-XS↑384 | 21 | 13.1 | 384 | 82.2 | 95.8 |
| SVT-S↑384 | 33 | 22.0 | 384 | 83.1 | 96.4 |
| SVT-B↑384 | 57 | 37.3 | 384 | 83.0 | 96.2 |

indicate that our SVT models outperform GFNet's base spectral network. For instance, SVT-S(384) achieves an impressive accuracy of 83.0%, surpassing GFNet-S(384) by 1.2%, as presented in Table 6. Similarly, SVT-XS and SVT-B outperform GFNet-XS and GFNet-B, respectively, highlighting the superior performance of our SVT models in the fine-tuning process.

## 3.5 Comparison with Similar architectures

We compare SVT with LiTv2 (Hilo) [15] which decomposes attention to find low and high-frequency components. We show that LiTv2 has a top-1 accuracy of 83.3%, while SVT has a top-1 accuracy of 85.2% with a fewer number of parameters. We also compare SVT with iFormer [19] which captures low and high-frequency information from visual data, whereas SVT uses an invertible spectral method, namely the scattering network, to get the low-frequency and high-frequency components and uses tensor and Einstein mixing respectively to capture effective spectral features from visual data. SVT top-1 accuracy is 85.2, which is better than iFormer-B, which is at 84.6 with a lesser number of parameters and FLOPS.

We compare SVT with WaveMLP [21] which is an MLP mixer-based technique that uses amplitude and phase information to represent the semantic content of an image. SVT uses a low-frequency component as an amplitude of the original feature, while a high-frequency component captures complex semantic changes in the input image. Our studies have shown, as depicted in Table- 3, that SVT outperforms WaveMLP by about 1.8%. Wave-VIT-B[25] uses wavelet transform in the key and value part of the multi-head attention method whereas SVT uses a scatter network to decompose high and low-frequency components with invertibility and better directional orientation using Einstein and Tensor mixing. SVT outperforms Wave-ViT-B by 0.4%.

## 3.6 SVT Compared with LVM/LLM

We wish to state the following on the comment of the reviewer about large vision models (LVM/LLM): We have observed in recent papers that certain efficient transformer models such as efficientFormer and CvT have a significantly larger number of parameters, with BiT-M having 928 million parameters and achieving 85.4% accuracy on ImageNet 1K, whereas ViT-H has 632 million parameters and achieving accuracy of 85.1. Comparatively, SVT-H-L has 54 million parameters and achieves 85.7% accuracy on ImageNet 1K - nearly 10X the lesser number of parameters and FLOPS but with improved accuracy, as captured in Table 3 of CvT [24].