# OpenReview forum: "Scattering Vision Transformer: Spectral Mixing Matters"
_NeurIPS.cc/2023/Conference — NeurIPS 2023 poster_

### Official Review · Reviewer_ghvx · 2023-06-26

**Soundness:** 3 good
**Presentation:** 2 fair
**Contribution:** 3 good
**Rating:** 7
**Confidence:** 3

**Summary:**

Assistant
The article is about the Scattering Vision Transformer, or SVT, which is a new adaptation of transformers for computer vision tasks. Its unique feature is the use of a spectrally scattering network, which captures fine-grained information about an image and addresses the issue of information loss caused by down-sampling operations. The SVT achieves state-of-the-art performance on the ImageNet dataset and outperforms other transformers in other vision tasks.

**Strengths:**

1. The use of DT-CWT, which is both shift-invariant and free from aliasing, is a robust and reliable approach.
2. The combination of tensor multiplication in low-frequency components and Einstein multiplication in high-frequency components is a compelling and efficient technique.
3. The experimental results demonstrate great promise for the proposed method.

**Weaknesses:**

1. To enhance comprehension, a brief illustration of DT-CWT through either descriptive language or illustrative figures should be provided in the method or background section.
2. Not all layers of vision transformers are scatter layers. An explanation and study should be provided.
3. The paper would benefit from the inclusion of mathematical theorem proofs. I found it difficult to understand the benefits of DT-CWT, particularly the decoupling of low and high frequencies. The paper's methods read more like a technical report than a compelling story of motivation, possibly due to the omission of some logistic or preliminary details.
4. To further evaluate the performance, provide comparisions of speed and over-fitting tests on ImageNet V2.
5. There are some typos in the paper. For example, on line 260, "figure ??" is not clear. Additionally, the official reference uses brackets [xx], whereas parentheses are used in the text. Please ensure consistency and accuracy in all references.

**Questions:**

1. Could you provide an explanation for why scatter layer is used in shallower layers but traditional attention layer is used in deeper layers?
2. Is there any evidence, such as references, mathematical induction, or related research studies, to support the idea of using tensor multiplication for low pass and Einstein multiplication for high pass?

**Limitations:**

There appear to be no potential negative social impacts.

---

> ### Author Rebuttal · Authors · 2023-08-09
>
> We thank the reviewer for the detailed comments, which we believe are insightful and shall help in improving the quality of the final submission.
>
> 1. A background section on DT-CWT [46] shall be added to the final version of the paper to help readers understand it, before explaining SVT method details. This has now been included in **the main rebuttal**.
>
> 2. We compare the performance of SVT when the architecture changes from all attention ( PVTv2[58] ) to all spectral layers ( GFNet[44] ) well as a few spectral and remaining attention layers(SVT ours). We observe that combining spectral and attention boost the performance compared to all attention and all spectral layer-based transformer as shown in Table-2 of the main paper.
>
> 3. We provide a mathematical characterization of DT-CWT [46]  below and how it helps in decoupling low and high-frequency components. We will include this in the background section of the paper.
>
> 4. We have conducted an experiment to measure the latency of SVT and compare the same with ViT and GFNet, the results of which are given in the Table-2 of the attached PDF.  This is shown in Table 2 below, where we compare latency, FLOPS, number of parameters, and reconstruction loss. The table-2 shows the Latency (mili sec) of SVT compared with the Convolution type network, attention type transformer network, POOl type transformer network, MLP-mixer type transformer network, and Spectral type transformer network. We report latency per sample on A100 GPU.
>
> 5. We have also conducted an experiment to test robustness by using the ImageNet C dataset, the results of which are reproduced in table-7 of the attached PDF. We also use an SVT pre-trained on the ImageNet-1K dataset and use the ImageNet-P dataset to measure the robustness of SVT compared with various transformers such as ViT-B, MLP-Mixer, and SVT as shown in the table-7 of the attached PDF. The table compares accuracy and mCE score on the ImageNet-C dataset. SVT-H-B has less mCE score compared to other transformers.
>
> 6. Thanks for pointing out typos etc. in the paper, we shall ensure a thorough revision of the paper and make sure to address all language issues.
>
> 7. The ablation study was conducted to show that initial scatter layers followed by attention in deeper layers are more beneficial than having later scatter and initial attention layers ( SVT-H-S-Inverse). We also compare transformer models based on an alternative the attention and scatter layer (SVT-H-S-Alternate). The results are documented in the table-3 of the attached PDF. From all these combinations we observe that initial scatter layers followed by attention in deeper layers are more beneficial than others.
>
> 8. The use of tensor multiplication for low pass and Einstein multiplication for high pass is our contribution. It must be noted that low-frequency components contain the energy component of the signal which requires all the frequency components to provide energy compaction, while high-frequency components can be represented by only a few components, which can be achieved using Einstein multiplication. We have evaluated the same empirically and shown in Table 7 of the main paper that SVT_{TTEE} is the best performing compared to other alternatives. We found that using Einstein multiplication in channel and token mixing makes the transformer architecture more efficient in terms of FLOPS and no. of parameters, without compromising on accuracy. We have also revised the contributions section of the camera-ready paper to reflect this.
>
>
>
> ## Mathematical Formulation of DTCWT
>
> \begin{equation*}
>      x(t)=  \sum_{n=-\infty}^{\infty}c(n)\phi(t-n)  +\sum_{j=0}^{\infty}\sum_{n=-\infty}^{\infty}d(j, n)2^{j/2}\psi(2^{j}t-n).
> \end{equation*}
> where $c(n)$ is the scaling coefficients and $d(j,n)$ is the wavelet coefficients.
> \begin{equation*}
> c(n)  =\int_{-\infty}^{\infty}x(t)\phi(t-n)dt, \quad
> d(j, n) =2^{j/2}\int_{-\infty}^{\infty}x(t)\psi(2^{j}t-n)dt.
> \end{equation*}
>
> The complex value wavelet is given by
> \begin{equation*}
> \psi_{\rm c}(t)=\psi_{\rm r}(t)+{\rm j}\psi_{\rm i}(t).
> \end{equation*}
> Where $\psi_r(t)$  is real and even and $j\psi_i(t)$ is imaginary and odd
> \begin{equation*}
> d_{\rm c}(j, n)=d_{\rm r}(j, n)+{\rm j}\ d_{\rm i}(j, n)
> \end{equation*}
>
> With Magnitude and phase
> \begin{equation*}
> \vert d_{\rm c}(j, n)\vert =\sqrt{[d_{\rm r}(j,n)]^{2}+[d_{\rm i}(j,n)]^{2}}, \quad
> \angle d_{\rm c}(j,n)= \arctan \left({d_{\rm i}(j,n)\over d_{\rm r}(j,n)}\right)
> \end{equation*}
>
> Let $h_0(n), h_1(n)$ denote the low-pass and high-pass filter in the upper band, while $g_0(n), g_1(n)$ denote the same for the lower band. The wavelets corresponding to the upper band and lower band are denoted by $\psi_h(n), \psi_g(n)$. The filters are designed to get the complex wavelet by satisfying the  Perfect Reconstruction (PR) conditions.
> The complex wavelet filter can be represented as $\psi(t):= \psi_h(t)+j\psi_g(t)$.
> where $\psi_g(t)$  is approximately the Hilbert transform of $\psi_h(t)$ i.e.  $\psi(t) \approx \mathcal{H}{\psi_h(t)}$ .
>
> \begin{equation*}
> \psi_{h}(t)=\sqrt{2}\sum_{n}h_{1}(n)\phi_{h}(t), \quad
>  \phi_{h}(t)=\sqrt{2}\sum_{n}h_{0}(n)\phi_{h}(t)
> \end{equation*}
>
>  Two low-pass filters should satisfy a very simple property: one of them should be approximately a half-sample shift of the other.
>
> $g_{0}(n)\approx h_{0}(n-0.5) \Rightarrow \psi_{g}(t)\approx {\cal H}\{\psi_{h}(t)\} $
>
> Since the filters are real, no complex arithmetic is required for implementing DTCWT. It is just two times more expansive in 1-D. It is also easy to invert, as the two separate DWTs can be inverted.
>
> Final Dual-Tree CWT can be designed using follow steps:
> - Approximate half-sample delay property
> - PR (orthogonal or biorthogonal)
> - Finite support (FIR filters)
> - Vanishing moments/good stopband
> - Linear-phase filters: Only the complex filter responses need to be linear-phase; this can be achieved by taking
>  $g_0(n)=h_0(N−1−n)$

---

> > ### Comment · Reviewer_ghvx · 2023-08-14
> > **Thank you for the response**
> >
> > The response has addressed my concerns, so I am updating my rating to "accept."

---

> > > ### Author Response · Authors · 2023-08-17
> > > **Replying to Reviewer ghvx**
> > >
> > > We thank reviewer ghvx for his insightful comments and shall revise the paper thoroughly to address his review comments.

---

### Official Review · Reviewer_upE5 · 2023-07-03

**Soundness:** 2 fair
**Presentation:** 3 good
**Contribution:** 2 fair
**Rating:** 4
**Confidence:** 4

**Summary:**

This paper proposed to use DTCWT in the transformer model in early stage, with the motivation of saving computation cost without information loss. The proposed scattering module has a low-pass and a high-pass branch. Tensor multiplication & Einstein multiplication are applied to the LF & HF branch respectively. The proposed module replaces the first 2 stages in hierarchical ViT models. The proposed SVT model achieved superior performance on image classification on the ImageNet1K dataset, and the downstream transfer learning results.

**Strengths:**

Replacing low level transformer modules with the proposed DTCWT based module achieves better accuracy and latency trade-off. Applying DTCWT to transformer models may not be explored yet by prior work.

**Weaknesses:**

DTCWT has been used in the previous CNN based deep learning method, e.g. Uses of Complex Wavelets in Deep Convolutional Neural Networks.
Existing work already shows that replacing early transformer stage with convolutional layers could give better accuracy and latency trade-off.

**Questions:**

How do we compare SVT with combining ViT with CNN, i.e. an architecture with CNN in early stages while transformer in late stages?

**Limitations:**

The proposed module is a highly hand engineered component. Adding manually designed components to the deep learning models may contradict with the motivation of removing inductive bias to the model and building generic learning based architectures.

---

> ### Author Rebuttal · Authors · 2023-08-09
>
>
> 1. **Comparing SVT with ViT with initial CNN layers:** We have conducted an experiment where the initial layers of a ViT are convolutional networks and later layers are attention layers to compare the performance of SVT. The results are captured in Table 4 in the attached PDF, where we compare SVT  with transformers having initial convolutional layers such as CVT, CMT, and HorNet. Initial  Convolutional layers in a transformer are not performing well compared to the initial scatter layer.    Initial scatter layer-based transformers are better performance and less computation cost compared to  Initial  Convolutional layer-based transformers which is shown in Table 4. We also measure the latency of various transformers as shown in Table 2. In this table, we compare latency, FLOPS, number of parameters, and reconstruction loss. The table-2 shows the Latency (mili sec) of SVT compared with the Convolution type network, attention type transformer network, POOl type transformer network, MLP type transformer network, and Spectral type transformer network. We report latency per sample on A100 GPU.
>
> 2. SVT is a generic recipe for componentizing the transformer architecture and efficiently implementing transformers with lesser parameters and computational complexity with the help of Einstein multiplication. So, this can be viewed as a simple and efficient learning transformer architecture with minimal inductive bias. We provide the source code for SVT in the paper and the reviewer is free to verify the above claim.

---

### Official Review · Reviewer_kC2p · 2023-07-06

**Soundness:** 3 good
**Presentation:** 1 poor
**Contribution:** 3 good
**Rating:** 6
**Confidence:** 4

**Summary:**

This paper proposes to use Dual-time Complex Wavelet Transforms to decompose images into high and low frequency components in vision transformers. With this technique, it claims to address the problem of attention complexity without the loss of information as in Fourier or DWT based Transformers. This claim is based on the invertible property of Dual-time Complex Wavelet Transforms. This paper claims to achieve state-of-art image classification performance on the ImageNet dataset with a significant reduction in a number of parameters and FLOPS, and comparable results in instance segmentation. Qualitative and quantitative results are provided.

**Strengths:**

1. The most attractive of this paper is the usage of Dual-time Complex Wavelet Transforms. The reviewer appreciates that the authors noticed the information loss in previous Fourier or DWT based Transformers, and proposed to adopt Dual-time Complex Wavelet Transforms to solve such a problem. It is invertible (though also redundant) and hence would not lead to a loss.

2. The paper provides a lot of technical details, and also the codes to reproduce the results.

3. Extensive experiments are conducted in the paper.

**Weaknesses:**

1. Clarity of Technical Contribution

The paper's presentation raises some concerns about clarity and comprehension. The key technical contribution of the work is obscured by convoluted explanations and distracting statements.

For example, L81-L93, at the beginning of the method section, this paper detailedly talks about something like "In Vanilla SVT, Given an image, we split it into patches of size 16*16. We use a linear  projection layer to get the embedding feature for each patch ......" These sentences seem to depict unique contributions of the proposed method but, upon closer inspection, merely explain standard operations in the Vision Transformer (ViT). Similarly, lines 102-109 are bewildering, appearing repetitive and unclear about the authors' intended message.

Moreover, the paper repeatedly emphasizes "tensor multiplication" and "Einstein multiplication" across various sections. These appear to be mere implementation details, not substantial technical contributions.

The manuscript's current style is closer a technical report rather than an academic paper. The authors are strongly encouraged to (a) explicitly state the key contribution of this work in the rebuttal and method section, and (b) condense lines 81-126 into a succinct, high-level introduction of the method, ideally within 10 lines.

2. Claims need support

Some of the claims in the paper, while potentially plausible, lack sufficient support/proof. For example, L52, "the ability to separate low-frequency and high-frequency components of an image is also important". In L100, "SVT has improved robustness compared to most other transformers, which will also be established in the performance studies." These could be true, but the authors did not prove the robustness is improved.

3. Minors

There are some minor typos/problems to be fixed. For example,  L130, "Where" to "where". In page 7, some table captions are bold and some are not.




**Questions:**

Overall the reviewer thinks using DTCWT in vision transformers is worth exploring, and this is why the reviewer still vote for borderline accept at this stage. If the authors cannot provide appropriate explanation/discussion about their contribution in the rebuttal period, the reviewer would downvote the score.

Besides, the reviewer feel some details provided by the main paper are not interesting/exciting, e.g., Table 1. For future work, the reviewer would suggest the authors to explore more insightful things. For instance, DTCWT enjoys invertibility but has redundancy at the same time. How would the network deal with such a redundancy? Moreover, providing more quantitive or qualitative comparisons to Fourier or DWT based Transformers to analyse how the invertibility of DTCWT helps vision transformers understand visual contents, instead of just reporting the final numbers for the tasks like classification.

____________________________________________________


The author response solves the main concerns. Under the assumption that the authors would revise the manuscript as they promised, the reviewer agreed to raise the score to weak accept.

---

> ### Author Rebuttal · Authors · 2023-08-09
>
> We thank the reviewer for the insightful comments, which shall result in significant improvements to the quality of the final camera-ready version of the paper
>
> 1. One of the claims of the paper is the ability of DTCWT [46] to separate the low-frequency and high-frequency components of the image. This has now been included in the background section of the paper and is captured in **the main rebuttal section**.
>
> 2. **Invertibility vs redundancy trade-off:** Regarding the invertibility vs redundancy trade-off, we conduct an experiment to show that invertibility helps in comprehending the image and not just contributing to the performance. We pass an image through raw DTC-WT and we put an inverse DTC-WT operation and compute the reconstruction loss.  We ran the experiment with various values of J, which represent the orientation in SVT. We observe that the reconstruction loss in the image reduces with increasing values of J, which clearly shows that as the orientations increase, SVT is able to comprehend the image better – the orientations are able to represent the high-order properties of the image which benefits SVT. We have compared different spectral transforms such as Fourier Transform, Discrete Wavelet Transform as well as DTC-WT. We observe that the reconstruction loss is lesser in DTC-WT as compared to other spectral transforms, as captured in Table 1 below.
> In Table 1 we measured reconstructed loss(MSE) for FFT, DWT stage -1,2,3, and DTCWT stage -1,2,3. It shows that the MSE loss is less as we increase the level of decomposition (J) and order of selectivity. DTCWT has less MSE than DWT. Similarly, the PSNR value of DTWT is gier than DWT and Fourier. The Peak Signal-to-Noise Ratio (PSNR) is a measure of the ratio between the maximum possible power of an image and the power of corrupting noise that affects its representation. It is defined via the MSE and is expressed in decibels (dB). The higher the PSNR, the better the quality of the reconstructed image. In summary, for a reconstructed image to be considered of high quality, it should have a low MSE and a high PSNR.
>
>  3. It must be noted that Fourier Transforms cannot perform low-pass and high-pass separation, as mentioned in the GFNet paper.  GFNet always uses only tensor multiplication, which may be inefficient compared to Einstein multiplication, which is efficient and reduces the number of parameters and computational complexity. Thus, while there is offset and SVT does not lag in performance nor in computational complexity but gains better representational power. This is shown in Table 2, where we compare latency, FLOPS, number of parameters, and reconstruction loss.  The table-2 shows the Latency (mili sec) of SVT compared with the Convolution type network, attention type transformer network, POOl type transformer network, MLP type transformer network, and Spectral type transformer network. We report latency per sample on A100 GPU. We have also visualized the filter coefficients of all six orientations in Supplementary Figure 1.
>
>  4. **Robustness:** We also use an SVT pre-trained on the ImageNet-1K dataset and use the ImageNet-C dataset to measure the robustness of SVT compared with various transformers such as ViT-B, MLP-Mixer, and SVT as shown in Table 7 of the attached PDF. The table compares accuracy and mCE score on the ImageNet-C dataset.  SVT-H-B has less mCE score compared to other transformers.
>
> 5. The quality of writing will be improved considerably in the final version and we shall also include a Mathematical Formulation of DTCWT.
>
> ## Mathematical Formulation of DTCWT
>
> \begin{equation*}
>      x(t)=  \sum_{n=-\infty}^{\infty}c(n)\phi(t-n)  +\sum_{j=0}^{\infty}\sum_{n=-\infty}^{\infty}d(j, n)2^{j/2}\psi(2^{j}t-n).
> \end{equation*}
> where $c(n)$ is the scaling coefficients and $d(j,n)$ is the wavelet coefficients.
> \begin{equation*}
> c(n)  =\int_{-\infty}^{\infty}x(t)\phi(t-n)dt, \quad
> d(j, n) =2^{j/2}\int_{-\infty}^{\infty}x(t)\psi(2^{j}t-n)dt.
> \end{equation*}
>
> The complex value wavelet is given by
> \begin{equation*}
> \psi_{\rm c}(t)=\psi_{\rm r}(t)+{\rm j}\psi_{\rm i}(t).
> \end{equation*}
> Where $\psi_r(t)$  is real and even and $j\psi_i(t)$ is imaginary and odd
> \begin{equation*}
> d_{\rm c}(j, n)=d_{\rm r}(j, n)+{\rm j}\ d_{\rm i}(j, n)
> \end{equation*}
>
> With Magnitude and phase
> \begin{equation*}
> \vert d_{\rm c}(j, n)\vert =\sqrt{[d_{\rm r}(j,n)]^{2}+[d_{\rm i}(j,n)]^{2}}, \quad
> \angle d_{\rm c}(j,n)= \arctan \left({d_{\rm i}(j,n)\over d_{\rm r}(j,n)}\right)
> \end{equation*}
>
> Let $h_0(n), h_1(n)$ denote the low-pass and high-pass filter in the upper band, while $g_0(n), g_1(n)$ denote the same for the lower band. The wavelets corresponding to the upper band and lower band are denoted by $\psi_h(n), \psi_g(n)$. The filters are designed to get the complex wavelet by satisfying the  Perfect Reconstruction (PR) conditions.
> The complex wavelet filter can be represented as $\psi(t):= \psi_h(t)+j\psi_g(t)$.
> where $\psi_g(t)$  is approximately the Hilbert transform of $\psi_h(t)$ i.e.  $\psi(t) \approx \mathcal{H}{\psi_h(t)}$ .
>
> \begin{equation*}
> \psi_{h}(t)=\sqrt{2}\sum_{n}h_{1}(n)\phi_{h}(t), \quad
>  \phi_{h}(t)=\sqrt{2}\sum_{n}h_{0}(n)\phi_{h}(t)
> \end{equation*}
>
>  Two low-pass filters should satisfy a very simple property: one of them should be approximately a half-sample shift of the other.
>
> $g_{0}(n)\approx h_{0}(n-0.5) \Rightarrow \psi_{g}(t)\approx {\cal H}\{\psi_{h}(t)\} $
>
> Since the filters are real, no complex arithmetic is required for implementing DTCWT. It is just two times more expansive in 1-D. It is also easy to invert, as the two separate DWTs can be inverted.
>
> Final Dual-Tree CWT can be designed using follow steps:
> - Approximate half-sample delay property
> - PR (orthogonal or biorthogonal)
> - Finite support (FIR filters)
> - Vanishing moments/good stopband
> - Linear-phase filters: Only the complex filter responses need to be linear-phase; this can be achieved by taking
>  $g_0(n)=h_0(N−1−n)$

---

> > ### Comment · Reviewer_kC2p · 2023-08-16
> >
> > After carefully reviewing the rebuttal and considering feedback from other reviewers, I am inclined to raise the score. I hope the authors can address and refine the manuscript as recommended by the reviewers. The current writing distracts.

---

> > > ### Author Response · Authors · 2023-08-17
> > > **Replying to Reviewer kC2p**
> > >
> > > We thank reviewer kC2p for his insightful comments and shall revise the paper thoroughly to address his review comments.

---

### Official Review · Reviewer_GT1n · 2023-07-07

**Soundness:** 3 good
**Presentation:** 2 fair
**Contribution:** 3 good
**Rating:** 5
**Confidence:** 4

**Summary:**

The paper proposes SVT, a novel vision transformer model that addresses the challenges of attention complexity and capturing fine-grained information in images. SVT utilizes a spectral scattering network and the Dual-time Complex Wavelet Transforms (DTCWT) to decompose image features into low-frequency and high-frequency components. The paper also introduces an efficient feature mixing technique using Einstein multiplication in the high-frequency components and tensor multiplication in the low-frequency components. Experimental results demonstrate that SVT achieves state-of-the-art performance on the ImageNet dataset with reduced parameters and computational complexity.

**Strengths:**

1.The proposed SVT model presents an innovative approach to addressing attention complexity and capturing fine-grained information in images. The use of spectral scattering and DTCWT decomposition enables efficient representation and separation of frequency components.
2.The feature mixing technique using Einstein multiplication is a novel contribution that efficiently combines token and channel features, leading to improved performance.
3.The experimental results on the ImageNet dataset and other vision tasks demonstrate the superiority of SVT compared to existing vision transformers, such as LiTv2 and iFormer. The significant reduction in parameters and computational complexity further enhances the practicality and scalability of SVT.


**Weaknesses:**

1. The English writing in this paper needs to be carefully reviewed as there are several grammar errors. For example, line 79, lesser should be less, line 83, "Given" should be in lower case "given". And in Figure 4's illustration, some parts are missing. There are many more errors in your paper writing.
2. I find the approach in this paper to be interesting but overly tricky. As we have seen with LLMs, many highly technical improvements become less significant when supported by the large parameter size and data volume of such models. Moreover, in the experimental section, the performance improvement brought by SVT does not appear to be significant. From my perspective, simple and elegant techniques often yield more meaningful improvements compared to complex transformations.


**Questions:**

In the experiments, why not compare your method with more popular  CNN-based methods like Yolo  and Faster-RCNN?

**Limitations:**

Author has presented some limitations of the proposed methods and future plan.

---

> ### Author Rebuttal · Authors · 2023-08-09
>
> We thank the reviewer for his comments.
>
> 1. **Language:** Thanks for pointing out this – we shall undertake a thorough revision of the paper to address all language issues.
>
> 2. SVT is a generic recipe for componentizing the transformer architecture and efficiently implementing transformers with lesser parameters and computational complexity with the help of Einstein multiplication. So, SVT can be viewed as a simple and efficient learning transformer architecture, as opposed to viewing it as a complex transformation. The essence of SVT is the tensor multiplication in low-pass and Einstein multiplication in high-pass to get efficient transformer architecture. SVT leverages DT-CWT to get the phase information or high-frequency components, which is not possible in DWT.
>
> 2. We have conducted an experiment to compare SVT’s performance on object detection tasks compared to Mask-RCNN,  as well as Faster-RCNN – the results of which are tabulated in Table-6 in the attached PDF. This table provides performance results on COCO val2017 dataset for the downstream task. Here We compare  Faster-RCNN with the Mask R-CNN 1x [21] method. We have reported the bounding box (\emph{i.e.}, $AP^b$ )  for evaluation.  The $AP^b$ scores for SVT using Mask RCNN are better than Fster-RCNN. We shall include the comparison with Yolo in the final version.
>
> 3. **SVT Compared with LVM/LLM**: We wish to state the following on the comment of the reviewer about large vision models (LVM/LLM): We have observed in recent papers that certain efficient transformer models such as efficientFormer and CvT have significantly larger number of parameters, with BiT-M has 928 million parameters and achieving 85.4% accuracy on ImageNet 1K, whereas ViT-H has 632 million parameters and achieving accuracy of 85.1. Comparitively, SVT-H-L has 54 million parameters and achieves 85.7% accuracy on ImageNet 1K - nearly 10X lesser number of parameters and FLOPS but with improved accuracy, as captured in Table 3 of CvT paper reference [60] of our paper.

---

### Official Review · Reviewer_jzMY · 2023-07-09

**Soundness:** 3 good
**Presentation:** 3 good
**Contribution:** 3 good
**Rating:** 7
**Confidence:** 4

**Summary:**

The paper introduces the Scattering Vision Transformer (SVT), which utilizes a spectrally scattering network to capture fine-grained information in images and addresses the invertibility issue. SVT incorporates a novel spectral mixing technique using Einstein multiplication for efficient channel and token mixing. The approach achieves state-of-the-art performance on the ImageNet dataset, significantly reducing the number of parameters and FLOPS. It also demonstrates competitive results in other vision tasks, including transfer learning on standard datasets.

**Strengths:**

## Novelty

The use of scattering networks and Fourier-like frequency processing is novel and innovative. The paper addresses a significant problem related to the texture processing ability of vision transformers, showcasing desirable novelty and originality.

## Quality & Clarity

The paper is well-organized, providing clear preliminaries, assumptions, definitions, and solutions. The experiments are detailed and concrete.

## Significance

SVT effectively separates low-frequency and high-frequency image components while reducing computational complexity through the Einstein multiplication-based mixing technique. It achieves state-of-the-art performance on image classification and instance segmentation tasks and shows comparable results in object detection tasks.

**Weaknesses:**

The computational costs and complexity limit the number of directional orientations used in SVT. Currently, SVT employs six orientations, but increasing the number of orientations would capture more semantic information at the expense of higher computational complexity. Optimization possibilities should be explored. Additionally, SVT's performance in domains such as speech and NLP remains unexplored.

**Questions:**

None

---

> ### Author Rebuttal · Authors · 2023-08-09
>
> We thank the reviewer for his insightful comments on the paper and to guide us on the possible research directions.
>
> 1. Regarding the invertibility VS redundancy trade-off, we conduct an experiment to show that invertibility helps in comprehending the image and not just contributing to the performance. We pass an image through raw DTC-WT and we put an inverse DTC-WT operation and compute the reconstruction loss.  We ran the experiment with various values of J, which represent the orientation in SVT. We observe that the reconstruction loss in the image reduces with increasing values of J, which clearly shows that as the orientations increase, SVT is able to comprehend the image better – the orientations are able to represent the high-order properties of the image which benefits SVT. We have compared different spectral transforms such as Fourier Transform, Discrete Wavelet Transform as well as DTC-WT. We observe that the reconstruction loss is lesser in DTC-WT as compared to other spectral transforms, as captured in Table 1 below. We also capture the invertibility VS redundancy trade-off in table-5.
> In Table 1 we measured reconstructed loss(MSE) for FFT, DWT stage -1,2,3, and DTCWT stage -1,2,3. It shows that the MSE loss is less as we increase the level of decomposition (J) and order of selectivity. DTCWT has less MSE than DWT. Similarly, the PSNR value of DTWT is gier than DWT and Fourier. The Peak Signal-to-Noise Ratio (PSNR) is a measure of the ratio between the maximum possible power of an image and the power of corrupting noise that affects its representation. It is defined via the MSE and is expressed in decibels (dB). The higher the PSNR, the better the quality of the reconstructed image. In summary, for a reconstructed image to be considered of high quality, it should have a low MSE and a high PSNR.
>
> 2. Optimization possibilities – we shall explore a few optimization possibilities to reduce the redundancy in the orientations of SVT for future work and these include:
>   - The selection of relevant orientations which best capture image properties, instead of using all six orientations
>   - The use of symmetric and anti-symmetric pairs in the orientations – for instance, $15^\circ$ and $165^\circ$, as well as $45^\circ$, and $135^\circ$, are such pairs that capture similar image properties, which we can leverage to optimize the number of orientations and reduce redundancy.
>    - Use of pyramidal decomposition of orientation – the finer layer requires more orientation compared to the coarser layer.
>
> 3. We shall be exploring the applicability of SVT in speech data as it is also a spectral signal. We shall also be exploring NLP datasets for SVT.
>
> 4. We are adding visualization for filter characterization DT-CWT in terms of low-frequency and all 6 six directional components of high-frequency components as shown in Figure-1 and 2. Figure-1 shows the phase and magnitude of FFT and the low-frequency component of DTCWT, whereas Figure-2 shows the phase and magnitude of the high-frequency component of DTCWT. It clearly indicates that it captures six degrees such as $15^\circ$, $45^\circ$, $75^\circ$, $105^\circ$, $135^\circ$, and $165^\circ$. This can't be captured by FFT and DWT.

---

### Author Rebuttal · Authors · 2023-08-09

We thank all the reviewers for their constructive suggestions. We intend to incorporate the feedback to obtain an improved revision of our paper – we sincerely believe that the comments shall improve the quality of the paper significantly.

We provide clarifications for the points raised by the reviewers:

1. We have added a separate background section and included a detailed explanation of Dual-Tree Complex Wavelet Transform (DTCWT) as suggested by the reviewers. Here we have included a diagram in the attached PDF for your kind reference. The same shall be added to the camera-ready version of the paper.

   - The explanation also captures the ability of the DTCWT to separate low-frequency and high-frequency components of an image as well as the use of DTCWT in vision transformers.
   - We also give a mathematical characterization of DTCWT.

2. We have added several experiments to support & validate the claims made in the paper.

    - **Initial Convolutional Layers VS Scatter Layers**: These experiments include a comparison of SVT with transformer architectures having initial convolutional layers and final attention layers. This is now substantiated with detailed experiments, where we compare SVT with Hornet, CMT, and CVT. The new table is numbered Table 5 in the attached PDF.

    - **Efficiency of Einstein Multiplication based SVT Implementation**: We have also conducted a new experiment to measure the latency of SVT and compare it with the latency of other types of transformers. The table, which is captured in Table 2 in the attached PDF, shows that SVT has lower latencies in spite of having redundancies, which proves the efficiency of the Einstein multiplication in high-frequency components. SVT also has lower FLOPS and a lesser number of parameters due to the efficient implementation.

    - **Qualitative and quantitative study of Invertibility**: We have conducted an additional experiment to compare the invertibility of DTCWT with Fourier transform and DWT – we measure the reconstruction loss of the transformer based on all three and compare the same in Table 1 of the attached PDF.  We have also visualized qualitatively the invertibility property of DTCWT, Fourier, and DWT shown in Figure-1,2. Table 1 captures the invertibility studies.

    - **Invertibility VS Redundancy Trade-off**: We have analyzed the trade-off between invertibility and redundancy of SVT, where we were able to show that if we increase the number of orientations and number of stages, the reconstruction loss decreases, which is indicative of better invertibility at the cost of redundancy. The efficient implementation based on Einstein multiplication offsets the redundancy with improved performance with a reduced number of parameters and computational cost. We also show that SVT’s orientation better represents the higher-order properties of the image. This is captured in Table 6 of the attached PDF.

    - **Robustness**: We have also conducted an experiment to measure the robustness of SVT by pre-training on ImageNet 1K and comparing it with other architectures like ResNet, ViT, and MLP Mixer on the ImageNet C dataset. This is captured in Table 7 of the attached PDF and clearly demonstrates that SVT. SVT achieves better robustness compared to other transformer architectures.

    - **Initial Scatter VS Initial Attention**: We have conducted an additional experiment to show that SVT with initial scatter layers and deeper attention layers performs better than SVT having initial attention layers and deeper scatter layers. This is captured in Table 3, in the attached PDF.

    - **Faster RCNN comparison**: We have included the SVT VS Faster RCNN comparison in Table 4.

3. We have revised the paper significantly to take care of grammatical and language issues, as suggested by reviewers.

## Overview of DTCWT and Decoupling of Low & High Frequencies

Discrete Wavelet Transform (DWT) replaces the infinite oscillating sinusoidal functions with a set of locally oscillating basis functions, which are known as wavelets Kingsbury et al.[46]. Wavelet is a combination of low-pass scaling function $\phi(t)$ and a shifted version of a band-pass wavelet function known as $\psi(t)$. It can be represented mathematically as given below:
\begin{equation*}
     x(t)=  \sum_{n=-\infty}^{\infty}c(n)\phi(t-n)  +\sum_{j=0}^{\infty}\sum_{n=-\infty}^{\infty}d(j, n)2^{j/2}\psi(2^{j}t-n).
\end{equation*}
where $c(n)$ is the scaling coefficients and $d(j,n)$ is the wavelet coefficients.

Kingsbury et al.[46] have identified four issues in DWT including oscillations, shift variance, aliasing, and lack of directionality. One of the solutions to solve the above problems is the Complex Wavelet Transform (CWT). CWT is inspired by Fourier representation and has a complex-valued scaling function and complex-valued wavelet function, as given below:

$\psi_{\rm c}(t)=\psi_{\rm r}(t)+{\rm j}\psi_{\rm i}(t)$

CWT is a double redundant tight frame in 1-D, able to overcome the four shortcomings mentioned above. DTCWT is a specific redundant type of CWT, which is based on two Filter Bank (FB) trees. The DTCWT uses two real DWTs, with the first one giving the real part of the transform, while the second one gives the imaginary part. The two real DWTs use two different sets of filters, which are jointly designed to give an approximation of the overall complex wavelet transform and satisfy the Perfect Reconstruction (PR) conditions.

Let $h_0(n), h_1(n)$ denote the low-pass and high-pass filter in the upper band, while $g_0(n), g_1(n)$ denote the same for the lower band. The wavelets corresponding to the upper band and lower band are denoted by $\psi_h(n), \psi_g(n)$. The filters are designed to get the complex wavelet by satisfying the PR conditions. Since the filters are real, no complex arithmetic is required for implementing DTCWT. It is just two times more expansive in 1-D. It is also easy to invert, as the two separate DWTs can be inverted.

---

### Decision · Program_Chairs · 2023-09-21

**Decision:**

Accept (poster)

**Comment:**

Reviewers were enthusiastic about the paper and recognized a new, innovative approach to an important problem. Reviewers some minor point in their review which have been addressed in the rebuttal. We invite the authors to take these changes into account in the camera-ready version of the paper.